# Insights on the historical biogeography of Philippine domestic pigs and its relationship with continental domestic pigs and wild boars

John King N. Layos[1,2]*, Ronel B. Geromo[3], Dinah M. Espina[3], Masahide Nishibori[1,3]*

**1** Laboratory of Animal Genetics, Graduate School of Integrated Sciences for Life, Hiroshima University, Higashi-Hiroshima, Japan, **2** College of Agriculture and Forestry, Capiz State University, Mambusao, Capiz, Philippines, **3** Department of Animal Science, Visayas State University, Baybay City, Philippines

* johnkinglayos5@gmail.com (JKNL); nishibo@hiroshima-u.ac.jp (MN)

**Data Availability Statement:** All relevant data are within the paper and its Supporting Information files.

## Abstract

The Philippine archipelago was believed to have never been connected to the Asian continent even during the severe Quaternary sea-level drops. As a result, the history of domestic pig (*Sus scrofa*) dispersal in the Philippines remains controversial and must have some anthropogenic origin associated with human migration events. In this study, the context of origin, dispersal, and the level of genetic introgression in Philippine domestic pigs were deduced using mitochondrial DNA D-loop analysis altogether with domestic pigs and wild boar corresponding to their geographic origin. The results revealed considerable genetic diversity (0.900±0.016) and widespread Asian pig-ancestry (94.60%) in the phylogenetic analysis, with admixed European pig-origin (5.10%) harboring various fractions of ancestry from Berkshire and Landrace. The close genetic connection between the continental wild boars and domestic pigs present in the Philippine domestic pigs corroborates our hypothesis of a genetic signal that may be associated with the recently reported multiple waves of human migrations to the Philippines. The Haplogroup D7, reported to occur only in Indo-Burma Biodiversity Hotspots, included a high frequency of Philippine domestic pig haplotypes (54.08%), which poses an interesting challenge because its distribution is not consistent with the hypothesized migration route of Neolithic Austronesian-speaking populations. We detected the first Pacific Clade signature and ubiquitously distributed D2 haplotypes (Asian major) on several Philippine islands. The analyses of mismatch distribution and neutrality test were consistent with the Bayesian skyline plot which showed a long stationary period of effective population size. The population decline was consistent with the pronounced population bottleneck in Asian and European pigs during the interglacial periods of the Pleistocene. The results of this study will support the conservation strategies and improvements of economically important genetic resources in the Philippines.

**Funding:** This study was supported by the Japanese Government through the Ministry of Education, Culture, Sports, Science and Technology and partially by the Department of Science and Technology - Science Education Institute (DOST-SEI) Accelerated Science and Technology Human Resource Development Program (ASTHRDP) of the Philippines.

**Competing interests:** The authors have declared no competing interests exist.

## Introduction

The Eurasian wild boar (*Sus scrofa* L.), recognized as the ancestor of the domestic pig, is one of the most widely distributed mammals found throughout Eurasia, including South and East Asia, and extending to North Africa. This species was also introduced into the Americas, Australia and Oceania [1]. Because of its relationship with human settlement and movement, studies on the phylogeography of *S. scrofa* have provided significant evidence revealing both anthropological and biogeographical history [2]. From the perspective of molecular phylogeny on a larger geographical scale, wild boars are genetically divided into Asian and European clades [3–6], which have split during the Mid-Pleistocene 1.6–0.8 Ma ago [7]. Wild boars from East and Southeast Asia predominantly exhibit greater genetic variation than European wild boars in both mtDNA [4] and nuclear markers [8]. Island Southeast Asia (ISEA) and mainland Southeast Asia (MSEA), known to be the phylogenetic origin of wild boar, is a biodiversity hotspot where most other species in the genus *Sus* occur [9]. Recently, seven major clusters of pig mitochondrial control region sequences were hypothesized to reflect domestication from genetically distinct subpopulations of wild boar [4]; D1 (European major), D2 (Asian major), D3 (India), D4 (Italy), D5 (Myanmar), and D6 (Pacific Clade) with the addition of the mitochondrial Southeast Asian haplogroup (MTSEA; referred herein as the D7 haplogroup), which is thought to be restricted to the Indo-Burma Biodiversity Hotspots (IBBH) [10]. Furthermore, the clear phylogenetic structure of these wild boars [3, 4], as well as their possible association with the hypothesized Neolithic expansion [4], has been well documented.

The Philippines is one of the most biologically rich regions in the world, with an exceptionally high level of endemism for a country of its size. The Philippines have repeatedly been tagged as a global priority region for conservation–a top hotspot for both terrestrial and marine ecosystems [11–14]. It hosts four endemic wild pigs including the Philippine warty pigs (*Sus philippensis*), Visayan warty pigs (*Sus cebifrons*), Palawan bearded pig (*Sus ahoenobarbus*), and Mindoro warty pig (*Sus oliveri*), as well as one native species shared with Sundaic biogeographic region, the Bornean bearded pig (*Sus barbatus*) [15, 16]. Interestingly, the Philippine archipelago was not thought to be connected to the Asian continent even during the last ice ages [17]. Therefore, the ubiquitous Eurasian wild boar *S. scrofa* is not native to the Philippines because it was unable to reach the archipelago [15–19] and was probably introduced as a domestic animal within the last few thousand years [19]. At present, the only potential domestic pigs identified in the archaeological record of the Philippines are from the Neolithic (4000–3000 cal. BP) and Early Metal Age (3000–2000 cal. BP) sites at Nagsabaran in Northern Luzon, which confirmed the clear distinction between the domesticated pig and the Philippine warty pigs [20, 21], which is associated with the Neolithic dispersal of Austronesian-speaking populations to ISEA and Oceania [4]. However, this has recently been questioned as there is no evidence of domestic pigs in Taiwan at a similarly early date, casting doubt on the possible Neolithic introduction of domestic pigs to the Philippines [22].

The Austronesian settlers first colonized the Philippine archipelago around 4,000 years ago [23, 24] and were believed to have initiated the dispersal and translocation of pigs in the country. However, the circumstantial lack of corroborating archaeological evidence supporting the introduction of pigs in the Philippines have long been daunting. Numerous archaeologists and geneticists have argued that connections with MSEA, as opposed to the "Out of Taiwan" model of dispersal are responsible for the introduction of the Austronesian languages and agriculture into ISEA [25, 26]. This overlapping hypothesis was further strengthened by the absence of Pacific clade signatures in both Taiwan and the Philippines. For this reason, Larson et al. [27] excluded the Philippines as a point of origin for domestic pigs further east in the Pacific. These haplotypes appear to have originated somewhere in peninsular Southeast Asia

and were transported via Malaysia, Sumatra, Java, and islands in Wallacea such as Flores, Timor, and the Moluccas [4, 27–29] and have shown strong support for linking Neolithic material culture between Vietnam and ISEA [20]. In contrast, some modern and ancient Philippine domestic pigs possessed a unique haplotype stemming from Lanyu Island (Orchid Island), which lies between the northern Philippines and southern Taiwan [30]. Despite this complex genetic evidence, molecular studies undertaken of domestic pigs in the Philippines are still very limited thus, the pattern of pig expansion and dispersal remains unclear. Therefore, the patterns of haplotype distribution observed in this study can be associated with the importance of identifying the prehistoric arrival of domestic pigs in the Philippines and it's spread across the islands, as rooted to the hypothesized migratory route of Neolithic Austronesian-speaking populations from Taiwan into the Philippines [19, 31] and the possible dispersal of pigs from MSEA through Palawan and the Sulu Archipelago.

To date, the only available information suggests that the Philippine domestic pig is a product of indiscriminate interbreeding between numerous domesticated endemic Philippine wild pigs and an introduced domestic pig breed [32] that has been able to survive and reproduce with minimal human intervention. However, this hypothesis remains daunting due to the insufficient molecular studies to support this claim, as the evolution and dispersal of Philippine domestic pigs has yet to be elucidated. Therefore, this study aims to describe the mtDNA variability, genetic structure and phylogeographic origin of Philippine domestic pigs, understanding if the currently observed genetic diversity and structure exhibit a signature of past demographic expansion relative to those observed in the MSEA, and finally to contribute relevant insights to the conflicting hypotheses that have been proposed to explain the introduction of pigs involving the Philippines.

## Materials and methods

### Sampling and laboratory analysis

In this study, blood samples and hair follicles were collected from Philippine domestic pigs (Fig 1A and S1 Table), in accordance with institutional, local, and national guidelines for animal handling and use in experiments established by the Laboratory of Animal Genetics,

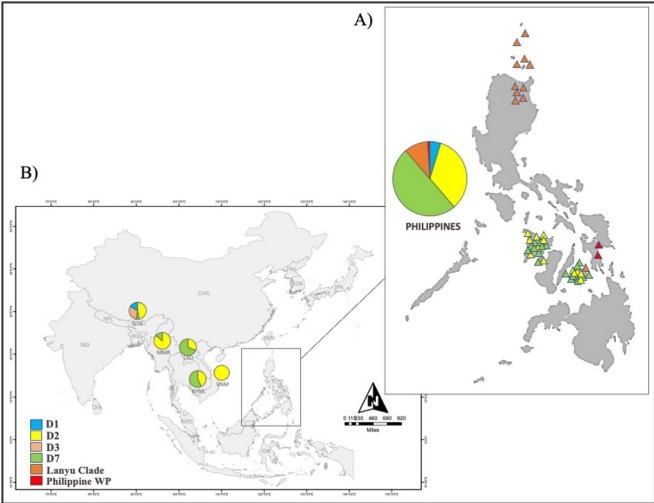

**Fig 1.** A) Geographic distribution of Philippine domestic pig haplogroups and the B) ancestry coefficients of mainland Southeast Asian pigs. Population names and abbreviations are detailed in S1–S3 Tables.

Hiroshima University (No. 015A170426). Our dataset also includes two samples of uncharacterized Philippine wild pigs confined at VSU, Visca, Baybay City, Leyte. The samples were preserved in tubes kept at -20˚C. Photographs were obtained to document the morphological characteristics and differences within these pig populations (Fig 2). The owners of the animals were personally consented to have their animals included in this study.

The genomic DNA was extracted using the phenol-chloroform method following the recommended protocol described by [33]. The 5.0-kbp mitochondrial DNA (mtDNA) fragment were amplified using a long and accurate–PCR kit (KOD FX-Neo Polymerase, TOYOBO, Otsu, Japan) using the established primer set: *Sus* mt. 5.0 FL-2: 5′–ATGAAAAATCATCGTTG TACTTCAACTACAAGAAC–3′; Mum R: 5′–TTCAGACCGACCGGAGCAATCCAGGTCGGT TTCTATCTA–3′. The reaction began with an initial denaturation at 94˚C for 2 min, followed by 30 cycles of denaturation at 98˚C for 10 sec, annealing at gradients at 57˚C for 30 sec, and primer extension at 68˚C for 2 min and 30 sec. The last step was 8 min final extension period at 68˚C. For mtDNA displacement (D-loop) region amplification, the ca. 1.3 kbp fragment was amplified using another primer set, *Sus* mtD F1: AACTCCACCATCAGCACCCAAAG, *Sus* mtD R1: CATTTTCAGTGCCTTGCTTTGATA. The reaction began with an initial denaturation at 94˚C for 2 min, after that, followed by 30 cycles of denaturation at 98˚C for 10 sec, annealing at gradients 59˚C for 30 sec, and extension at 68˚C for 30 sec. The last step was a 5 min final extension period at 68˚C. The amplification was done using the GeneAmp PCR System 9700 (Applied Biosystems, Foster City, CA, USA). The PCR products from the segmental amplification were cleaned and purified using Exonuclease I (ExoI) and Shrimp Alkaline Phosphatase (SAP) to degrade the residual PCR primers and dephosphorylate the remaining dNTPs, respectively. After this, samples were sequenced using ABI3130 sequencer for direct DNA sequencing and fragment analysis.

## Phylogenetic and population structure analysis

The profile alignments of the sequenced mtDNA D-loop data were performed using the CLUSTAL W algorithm implemented in Molecular Evolutionary Genetics Analysis (MEGA) [28]. About 1044 bp of the control region sequences were aligned and edited until one highly

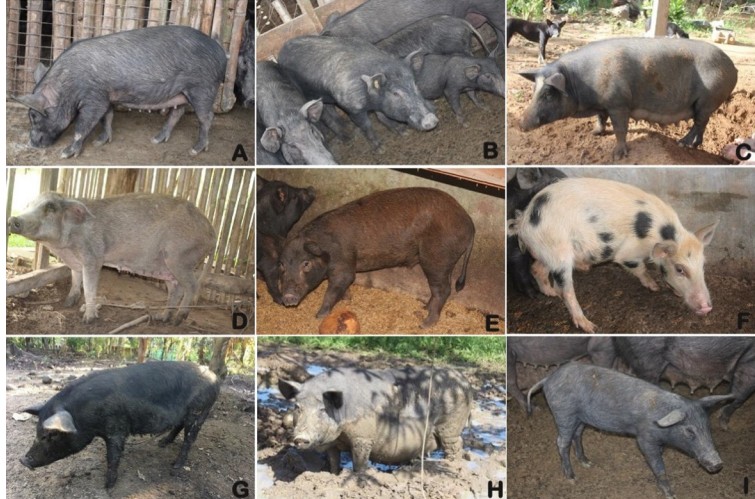

**Fig 2. A sample of the morphological variations across Philippine domestic pigs.** A) Albur, Bohol; B) Bugasong, Antique; C) San Miguel, Bohol; D) Barbaza, Antique; E) Balilihan, Bohol; F) Nueva Valencia, Guimaras; G) Talibon, Bohol; H) Dingle, Iloilo; I) Pilar, Bohol.

variable tandem repeat motif (5′-CGTGCGTACA-3′) remained. Haplotype sequences were submitted to GenBank National Center for Biotechnology Information (NCBI) databases (MN625805-MN625830; MW924902-MW92973). To place our results in a broader context, the sequences obtained in this study were truncated from the original size of 1044 bp to 510 bp to allow comparison of the pooled sequences with the previously documented sequences from MSEA available in GenBank (S2 Table).

The diversity measures such as the number of polymorphic segregating sites ($S$), haplotype diversity ($hd$) and nucleotide diversity ($\pi$) were estimated using DNA Sequence Polymorphism (DnaSP) 5.10 software [34]. The genetic structure was estimated employing the analysis of molecular variance (AMOVA) using Arlequin v.3.5.2.2 software [35] with 10,000 permutations. Four genetic structure hypotheses were tested based on geographical locations, (1) Philippine pigs (no groupings); (2) Philippines vs. MSEA combined; (3) Philippines *vs*. Bhutan *vs*. Myanmar, Laos, Cambodia, and Vietnam; (4) Philippines *vs*. Bhutan and Myanmar *vs*. Cambodia, Lao and Vietnam. $\Phi_{CT}$ is the difference among groups, $\Phi_{SC}$ is the difference among populations within groups, and $\Phi_{ST}$ is the difference within populations. The population pairwise net genetic distance based on population pairwise $F_{ST}$ (significant values were accepted at $p < 0.05$) was further computed using Arlequin. The level of significance was evaluated based on 1,000 random permutations. The indices of genetic structure estimated by AMOVA were determined using information on the allelic content of haplotypes and their frequencies [36], while the $F_{ST}$ was used to estimate the overall genetic divergence between populations.

All haplotypes were combined, including the downloaded sequences representing animals classified as domestic and wild *S. scrofa* from Europe and Asia (S3 Table), to construct the phylogeny using Bayesian Inference (BI) using the program Mr. Bayes 3.2 [37], with GTR+G+I as the best-fitted model determined by MEGA 7.0.26 [38] and jModelTest [39]. The Marcov chain Monte Carlo (MCMC) algorithm was run for 10,000,000 generations, with sampling every 1000 generations. The warthog (*Phacochoerus africanus*; DQ409327) was used as an outgroup. To obtained more detailed information on the phylogenetic relationship among these haplotypes, a reduced median network was constructed using Network v.4.1 [40, 41], available at http://www.fluxus-engineering.com. This method calculates the net divergence of each taxon from all other taxa as the sum of the individual distances from variance within and among groups. The nomenclatures described by Larson et al. [4] with six clades (D1 to D6) including the new clade proposed by Tanaka et al. [10] were used as the reference for clade notation.

## Population demographic analysis

Demographic history was inferred by analyzing the distribution of the number of site differences between pairs of sequences (mismatch distribution), performed on the pooled samples described previously, as implemented in DnaSP 5.10 software [34]. Expected values for a model of population growth-decline were calculated and plotted against the observed values. Populations that have experienced rapid demographic growth in the past show unimodal distributions, while populations that are in demographic equilibrium or decline show multimodal distributions [42]. Harpending's [43] raggedness index (H*ri*; quantifies the smoothness of the mismatch distributions and distinguishes between population expansion and stability) and sum of squared deviations (SSD) (1,000 simulated samples of pairwise nucleotide differences), as implemented in ARLEQUIN [43], were used to evaluate Rogers' [44] sudden expansion model which fits to a unimodal mismatch distribution [42]. To test for population expansion, we employed three additional tests: Fu's [45] $F_S$ and Tajima's *D* statistical tests using ARLEQUIN and testing their significance over 1,000 permutations; and Ramos-Onsin's and Roza's

$R^2$ test [39] using DnaSP. Statistical tests and confidence intervals for $F_S$ were based on a coalescent simulation algorithm and for $R^2$ on parametric bootstrapping with coalescence simulations.

The past population dynamics were also examined with the Bayesian Skyline Plot (BSP) [46] model with standard Markov Chain Monte Carlo sampling procedure (MCMC) under HKY + G model of substitution [47] with four gamma categories using Beast v.2.6.0 [48]. The BSP represents changes in population size over time derived from mtDNA and assumed mutation rate. Two independent analyzes were performed using all sequences from this study and the 130 sequences from MSEA using a mutation rate of $1.36 \times 10^{-8}$ (mutation rate per nucleotide site per year according to previous estimates for mammalian mtDNA control region) [49] under the strict clock. The MCMC analysis was performed for 50,000,000 generations. Independent runs (logs and trees) were pooled using Log Combiner, discarding the burn-in of the first 10% and sampling parameter values every 5,000 generations. Tracer v.1.7 was used to confirm the correct convergence of the MCMC chain with an effective sample size (ESS) > 200 within the log files and to visualize the dynamics of the effective population size over time. The light-blue shaded area marks the 95% highest posterior density (HPD). The X-axes are time in thousands of years before present (BP) and the Y-axes are the mean effective population size in millions of individuals divided by generation time on a logarithmic scale.

## Results

### Genetic diversity and population differentiation

Overall, 236 sequences including the 106 Philippine domestic pigs (two wild pigs were excluded) were used to measure population genetic structure and differentiation. The nucleotide sequences were aligned with respect to the representative haplotypes of Asian domestic pigs under accession number AB041480 (S1 Fig). In the alignment of sequences, all variable sites represented substitution mutation. A total of 23 haplotypes were detected from the Philippines (PHL), 11 in Cambodia, 9 in Bhutan, 10 in Laos, 17 in Myanmar, and 4 in Vietnam (Table 1). These sequences collapsed when pooled from 74 to 57 haplotypes. Twenty-six (6 PHL and 20 MSEA) of the 57 haplotypes were found only once among the sequences. The highest number of shared individuals was noted in PH37 haplotype, which consists of 31 sequences (25 PHL and 6 MSEA; 13.59%), corresponding to D7 haplogroup and previously referred to as the mitochondrial Southeast Asian haplogroup (MTSEA) [10]. The combined haplotype diversity of domestic pigs in MSEA was 0.966±0.006, with Myanmar having the

**Table 1. mtDNA indices of Philippine domestic pigs and mainland Southeast Asian pigs.**

| COUNTRIES | n | vs | P | S | #h | MPD | hd | π |
|---|---|---|---|---|---|---|---|---|
| KHM | 49 | 13 | 9 | 4 | 11 | 4.272±2.154 | 0.854±0.028 | 0.008±0.005 |
| BTN | 30 | 16 | 9 | 7 | 9 | 4.855±2.436 | 0.807±0.051 | 0.010±0.006 |
| LAO | 16 | 12 | 8 | 4 | 10 | 3.867±2.049 | 0.925±0.047 | 0.008±0.005 |
| MMR | 29 | 18 | 11 | 7 | 17 | 4.502±2.282 | 0.958±0.018 | 0.009±0.005 |
| VNM | 6 | 4 | 2 | 2 | 4 | 2.000±1.304 | 0.800±0.172 | 0.004±0.003 |
| TOTAL | 130 | 37 | 23 | 14 | 44 | - | 0.966±0.006 | 0.010±0.006 |
| PHL | 106 | 29 | 25 | 4 | 22 | 5.207±2.542 | 0.900±0.016 | 0.012±0.006 |

PHL = Philippines; KHM = Cambodia; BTN = Bhutan; MMR = Myanmar; VNM = Vietnam; *n* = number of samples; vs = variable sites; P = parsimony informative sites; *S* = singleton variable sites; #h = number of haplotypes; MPD = mean pairwise differences (SD); *hd* = gene diversity: haplotype level (SD); π = nucleotide diversity (SD).

highest haplotype diversity (0.958±0.018), followed by Laos with 0.925±0.047 and the Philippines with 0.900±0.016. The lowest haplotype diversity was observed from Vietnam (0.800 ±0.172). On the other hand, nucleotide diversity ($\pi$) was highest in the Philippines (0.012 ±0.006), closely similar to Bhutan (0.010±0.006), while the lowest was noted in Vietnam pigs (0.004±0.003). However, the small sample size of pigs from Vietnam could be the reason for the apparent low genetic diversity.

Based on the AMOVA of mtDNA D-loop data, all hypotheses subjected for analysis revealed a significant population subdivision ($\Phi_{ST}$ values, $p<0.01$), suggesting distinct genetic structure indicating significant genetic structure among geographic locations examined (Table 2). The inference of genetic differentiation was equally evident in both the BI and network analyzes (Figs 3 and 4). The pairwise divergence ($F_{ST}$ values) estimate ranged from 0.050 to 0.476 (Table 3A) and showed significantly higher genetic differentiation among most pigs in the population studies and between Philippine islands. However, insignificant and small pairwise $F_{ST}$ estimates were observed in domestic pigs from Laos and Cambodia ($F_{ST} = 0.050$; $p = 0.0631$), indicating that pigs from these countries were not isolated from each other consistent with the findings of Tanaka et al. [10]. The corrected average pairwise differences revealed consistent tendencies with the pairwise $F_{ST}$ (Table 3B).

To investigate the relationship of Philippine domestic pigs from those Asian and European domestic pigs, we included 40 sequences [4] to accommodate most of the major porcine mtDNA haplotypes. A total of 285 sequences were used to perform the phylogenetic tree and median-joining network analysis. The Bayesian phylogenetic tree branched into two core lineages, one of Asian phylogeographic origins and one of European phylogeographic origins (Fig 3). Network analysis generally supported the phylogenetic tree and revealed strong genetic structuring among Philippine pig haplotypes, where different phylogroups could be observed (Fig 4). Widespread Asian ancestry (94.90%) was observed in all Philippine domestic pig haplotypes examined. Seven (PHL1, 5, 7, 10, 14, 16 and 18) of 21 Philippine pig haplotypes nested under the D7 haplogroup, which accounted for 49.07% of the total population studied. This haplogroup was previously described in MSEA as restricted to Indo-Burma Biodiversity Hotspots (IBBH) [10] and as a distinct clade not described in previous studies [4]. Eight haplotypes (PHL2, 3, 4, 6, 8, 9, 11 and 12; 36.73%) were distributed in the D2 haplogroup, which corresponds to what Scandura et al. [5] recognized as widely distributed Chinese domestic pigs, a

**Table 2. Analysis of molecular variance (AMOVA) of Philippine domestic pigs and mainland Southeast Asian pigs.**

| HYPOTHESIS | Sources of variation | % Variation | $\Phi$ |
|---|---|---|---|
| PHL pigs (No groupings) | Within populations | 16.95** | $\Phi_{ST} = 0.16952$ |
| | Among populations within groups | 83.05 | $\Phi_{SC} = 0.83052$ |
| PHL vs. MSEA combined | Among groups | -9.34 | $\Phi_{CT} = -0.09338$ |
| | Among populations within groups | 21.47** | $\Phi_{SC} = 0.19639$ |
| | Within populations | 87.87** | $\Phi_{ST} = 0.12135$ |
| PHL vs. BTN vs. MMR, VNM, KHM, LAO | Among groups | 0.01 | $\Phi_{CT} = 0.00006$ |
| | Among populations within groups | 14.61** | $\Phi_{SC} = 0.14614$ |
| | Within populations | 85.38** | $\Phi_{ST} = 0.14619$ |
| PHL vs. BTN, MMR vs. VNM, KHM, LAO | Among groups | 2.51 | $\Phi_{CT} = 0.14837$ |
| | Among populations within groups | 12.33** | $\Phi_{SC} = 0.12644$ |
| | Within populations | 85.16** | $\Phi_{ST} = 0.14837$ |

**$p<0.01$ as tested by randomization (1000 permutations) using Arlequin; MSEA = Mainland Southeast Asia; PHL = Philippines; KHM = Cambodia; BTN = Bhutan; MMR = Myanmar; VNM = Vietnam.

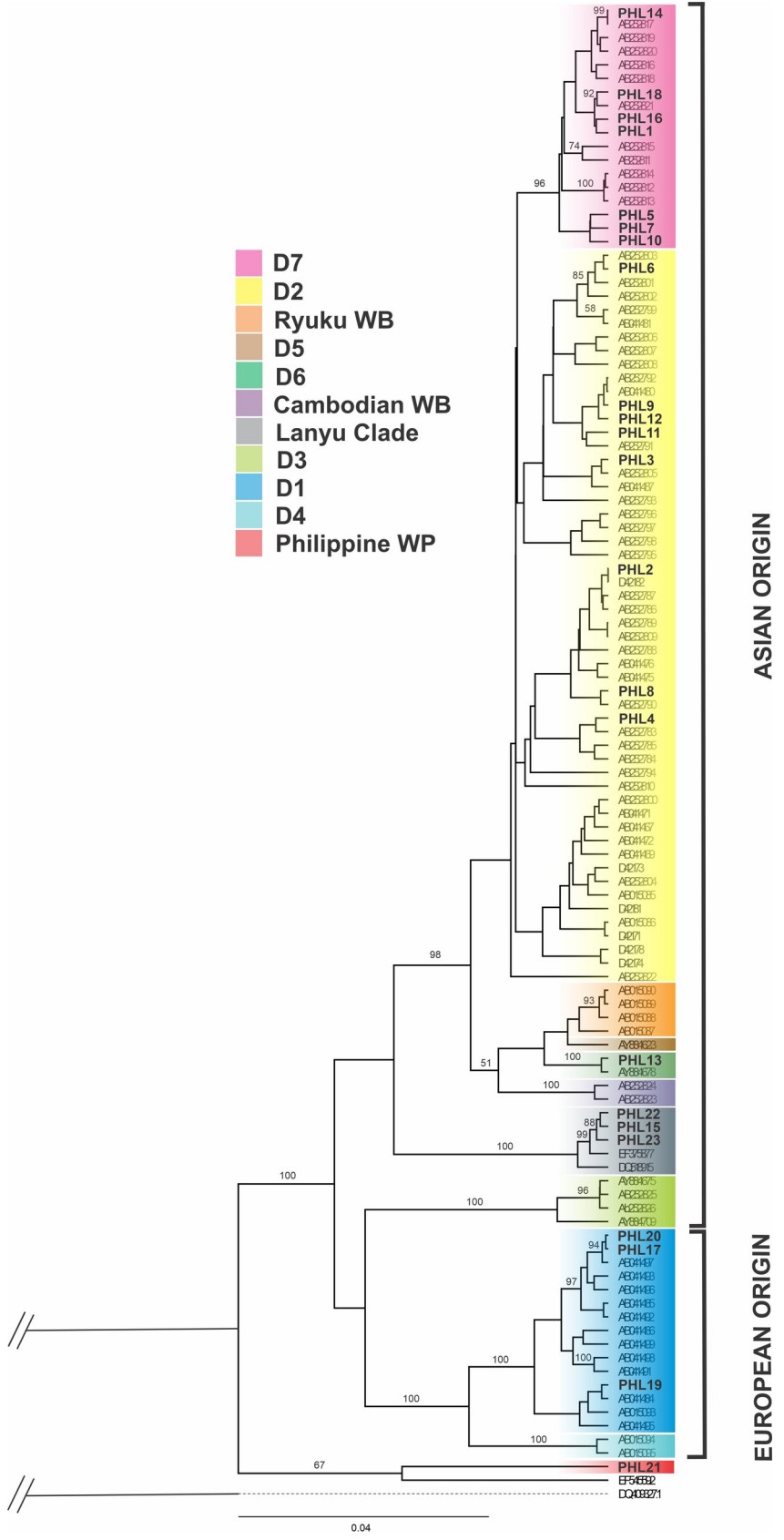

**Fig 3. Phylogenetic relationships of Philippine domestic pigs and wild pigs with continental domestic pigs and wild boar.** The number indicated in the nodes were probability support with warthog as the outgroup. Probability supports lower than 50% were not shown. Philippine domestic pigs revealed to comprised founder sources from five different geographic origins excluding the endemic Philippine wild pigs.

global pig breed that has some relationship with Asian pigs, as well as with East Asian wild boars [50, 51]. Three haplotypes (PHL17, 19 and 20; 5.10%) revealed the presence of mixed ancestry in the D1 haplogroup (European clade), harboring different fractions of maternal lineages from Berkshire and Landrace. Interestingly, one and three haplotypes, respectively, were assigned to the Pacific Clade (PHL13) and the distinct Type I Lanyu pig (PHL15, 22 and 23). As we shall discuss later, no similar haplotypes were found in previous studies on the existence of Pacific Clade haplotypes in the Philippines. One haplotype (PHL21) of a Philippine wild pig cannot be classified under any of the proposed haplogroups and formed a vague cluster from the porcine mtDNA control region haplotypes found in the previous research. It possessed a

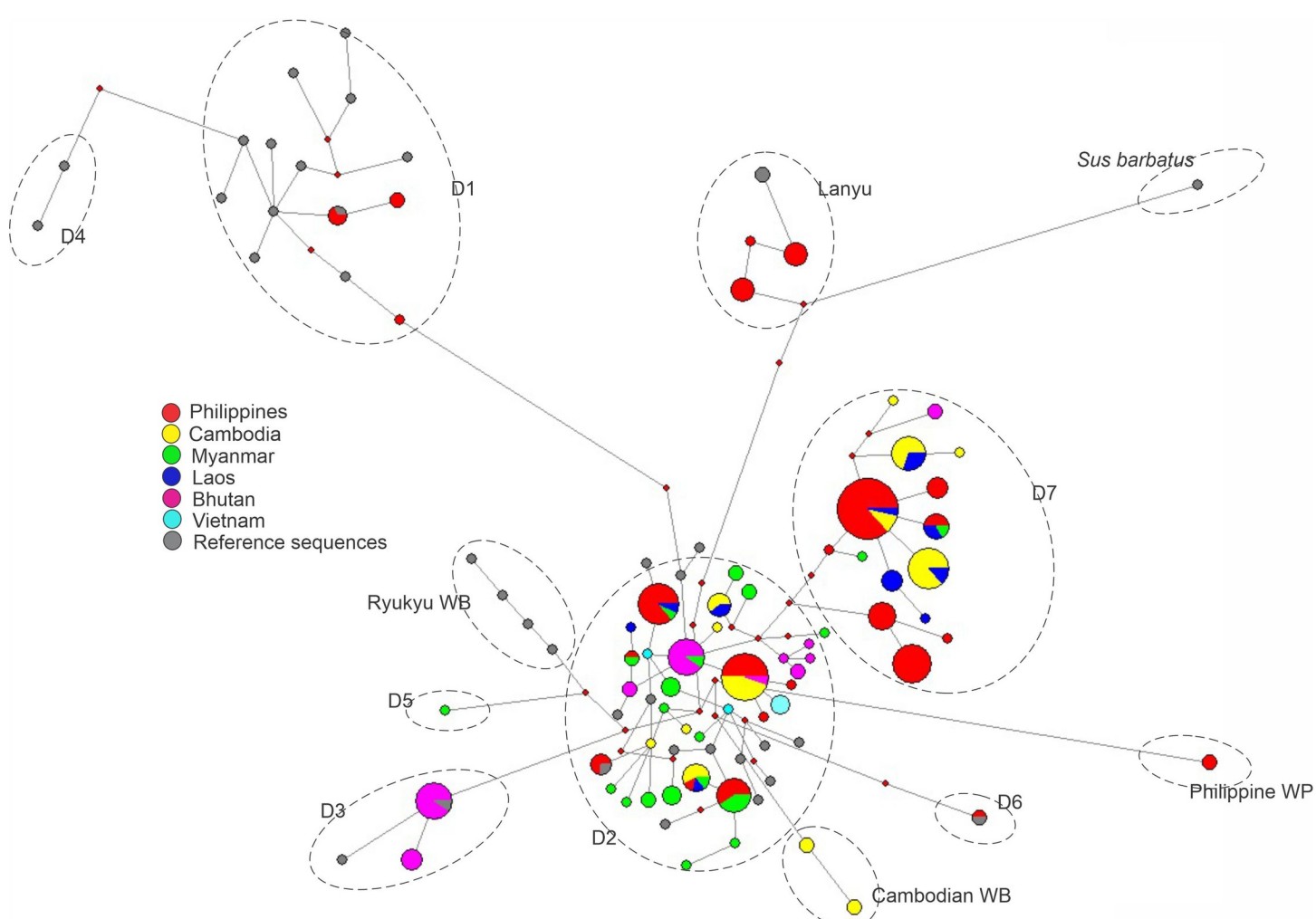

**Fig 4. The median-joining network of Asian and European pig haplotypes including the global reference sequences showing haplogroup classification.** Some haplotypes clustered together coinciding with their geographic area of origin, while selected haplotypes diverse and shared by individuals of different breeds from different geographical regions, indicating a negative correspondence between the geographic origin and the relationships among breeds. The size of each circle is proportional to the haplotype frequency. Color represents regions of sequence origin.

**Table 3. Genetic divergence among populations of Philippine domestic pigs and mainland Southeast Asian pigs.**

| (a) Population pairwise $F_{ST}$ | KHM | BTN | LAO | MMR | VNM | PHL |
|---|---|---|---|---|---|---|
| KHM | - | 0.301 | 0.050 | 0.169 | 0.280 | 0.061 |
| BTN | <0.0001 | - | 0.393 | 0.150 | 0.214 | 0.186 |
| LAO | 0.0721 | <0.0001 | - | 0.261 | 0.476 | 0.105 |
| MMR | <0.0001 | <0.0001 | <0.0001 | - | 0.203 | 0.086 |
| VNM | <0.0001 | 0.0091 | <0.0001 | <0.0001 | - | 0.120 |
| PHL | <0.0001 | <0.0001 | <0.0001 | <0.0001 | 0.0270 | - |
| (b) Population average pairwise difference | | | | | | |
| KHM | | 6.911** | 4.296 | 5.272** | 4.884** | 5.01 |
| BTN | 1.928** | | 7.296** | 5.508** | 4.833 | 1.673** |
| LAO | 0.227 | 2.935** | | 5.707** | 6.125** | 0.568** |
| MMR | 0.884** | 0.839** | 1.522** | | 4.471* | 0.704** |
| VNM | 1.748** | 1.406* | 3.192** | 1.22** | | 1.407* |
| PHL | 5.01 | 6.704** | 5.105 | 5.559 | 5.01 | |

(a) Population pairwise genetic distance. Upper triangular matrix: population pairwise estimates of $F_{ST}$; lower triangular matrix: the *p*-values for the $F_{ST}$; (b) average pairwise genetic differences between population. Upper triangle matrix: average number of pairwise differences between populations (PiXY); lower triangular matrix: corrected average pairwise difference [PiXY-(PiX + PiY)/2]. PHL = Philippines; KHM = Cambodia; BTN = Bhutan; MMR = Myanmar; VNM = Vietnam. *$p<0.05$; **$p<0.01$ as tested by randomization (1000 permutations) using Arlequin.

unique nucleotide polymorphism at sites T54C, C127T, A148G, T406C, G407A and a transversion substitution at the base G88T (S1 Fig).

## Past population dynamics

The mismatch distributions were also calculated to investigate the population expansion hypothesis. The distribution of pairwise nucleotide differences of the studied Philippine domestic pig populations revealed multimodal patterns of mismatch distribution (S2A Fig), which may suggest a population in decline or stable demographic equilibrium. In contrast, combined domestic pigs in MSEA revealed a unimodal mismatch distribution, a significant and large negative Fu's $F_S$ value (-25.282; $p<0.01$), along with the small and non-significant value of Harpending's raggedness index (H*ri*), likely supporting a scenario of demographic expansion experienced in the past (S2B Fig and Table 4). When the effects of natural selection

**Table 4. Values of neutrality test (Fu's $F_S$ and Tajima's *D*), sum of square deviation (SSD) and Harpending's raggedness index (H*ri*) for Philippine domestic pigs and MSEA pig mtDNA D-loop region.**

| COUNTRIES | Fu's $F_S$ test | Tajima's D test | SSD | H*ri* |
|---|---|---|---|---|
| KHM | 0.421 | 1.400 | 0.054* | 0.101* |
| BTN | 1.049 | 0.678 | 0.048 | 0.089 |
| LAO | -2.73 | 0.264 | 0.035 | 0.061 |
| MMR | -6.542* | -0.061 | 0.004 | 0.016 |
| VNM | -0.499 | 0.768 | 0.037 | 0.129 |
| TOTAL | -25.282** | -0.014 | 0.006 | 0.012 |
| PHL | -1.503 | 0.231 | 0.017 | 0.020 |

*$p<0.05$;
**$p<0.01$ as tested by randomization (1000 permutations) using ARLEQUIN. PHL = Philippines; KHM = Cambodia; BTN = Bhutan; MMR = Myanmar; VNM = Vietnam.

and past demographic changes are examined in each population by country using estimates of Tajima's $D$ and Fu's $F_S$, only Myanmar had a significantly negative Fu's $F_S$ values (-6.542; $p<0.05$), while all Tajima's $D$ values were not significant for all populations ($p = 0.05$; S3 Fig and Table 4). In addition, negative Fu's $F_S$ estimates were observed in domestic pigs in the Philippines (-1.503; $p = 0.05$), Laos (-2.73; $p = 0.05$) and Vietnam (-0.0499; $p = 0.05$). The Ramos-Onsin's & Roza's [52] $R^2$ tests were not significant in all cases except for the MSEA samples combined ($p<0.05$). The null hypothesis of expectation under the sudden expansion model, as indicated by the result of the sum square deviation and raggedness test result ($p = 0.05$) for the coalescence estimates, was not statistically supported in all locations examined, except for the combined MSEA samples. Analysis of prehistoric population size dynamics of Philippine domestic pigs using BSP was consistent with the result from the mismatch distribution analyzes. As projected by the BSP, the Philippine domestic pig population showed a long stationary period of effective population size (S4A Fig). The population decline occurred approximately 25,000 years before present (BP). On a regional scale, a gradual significant increase in population size was observed in MSEA domestic pigs during approximately the Late Pleistocene (S4B Fig).

## Discussion

Literature on genetic studies in Philippine domestic pigs is scarce, although this animal represents excellent genetic resources for the local economy and could serve as basis to study human settlements and migration. This study provides the first comprehensive data on the history of dispersal, genetic structure and diversity, and population dynamics of Philippine domestic pigs. The phylogenetic patterns in the current study were generally consistent with the two existing core lineages, one of Asian and one of European phylogeographic origin. The distribution of the haplotype frequencies did not indicate equilibrium thus, the geographically distributed haplotypes suggest that the present-day Philippine domestic pigs have multiple ancestors distributed across the Eurasian Continent. The close genetic connection between the continental wild boars and domestic pigs from the MSEA and Northeast Asia (NEA) present in the Philippine domestic pig genetic pool corroborates our hypothesis of a genetic signal that could potentially be associated with the recently reported multiple waves of human migrations to the Philippines during the last 50,000 years [53]. Extensive geneflow between *Sus* species is thought to have occurred during the ice ages [54] and is considered an important factor in the current geographic distribution of *Sus* populations around the world [55]. Thus, these events may have paved the way for the introduction of multiple distinct lineages of domestic pigs to the Philippines, as has been documented for chickens [56], goats [57], cattle [58], and other species that have adapted to local conditions and developed distinctive traits. In addition, the multifaceted history of rich trade and barter between travelers and coastal communities, including river movements in coastal settlements in the Philippines in prehistoric and proto-historic times [59], could be thought to have contributed significantly to the genetic landscape of the Philippines [53].

The complexity surrounding the Neolithic Austronesian expansion and dispersal has recently raised several questions due to the limited genetic studies conducted in the Philippine archipelago, as the faunal assemblages in this key region have provided some clues [4]. For instance, preliminary genetic studies revealed the absence of the Pacific Clade in the Philippine Archipelago [29] have led some researchers to challenge the veracity of the dispersal of domestic pigs from NEA into the Pacific islands via the Philippines [28]. However, in this study, we reported the first signature of the Pacific Clade and the ubiquitously distributed D2 haplotypes in the Philippines, which could potentially shed light on the issue of pig dispersal by the

Austronesian-speaking populations from NEA via the Philippines. In addition, the close genetic association of three haplotypes (PHL15,22–23) to Lanyu pigs from Taiwan strongly suggests that the maternal mtDNA originated from the same lineage.

Before the Europeans arrived in the Philippines during the Spanish colonization, there was evidence that domestic pigs had already been introduced by the Chinese traders [60], and subsequently followed by the intensive importation of various exotic pig breeds from Europe [61] resulting in a diversified genetic pool of Philippine domestic pigs. This hypothesis is supported by the close genetic relationships between Philippine domestic pigs and Chinese domestic pigs, which is exhibited by the similarities in their morphology and mtDNA variation due to introgression. It is undeniable that the Chinese mtDNA footprint has played an important role in the evolutionary history of the Philippine domestic pigs. Compared with the high signal of genetic introgression of Chinese pig breeds into Philippine domestic pigs, maternal introgression of European pigs was minimal, accounting for only 5.10% of the studied population. This observation is consistent with the situation in domestic pigs in the MSEA [10], where the mtDNA of European pigs has a negligible impact on the maternal origin of domestic pigs. Considering that our samples are aggregates from both lowland and upland areas, our visual observations and molecular result suggest that the exotic pig breeds have not yet fully invaded the remote areas of the Philippines. However, the widespread indiscriminate hybridization between exotic pig breeds and indigenous pigs (e.g., Berkshire or Duroc x Philippine domestic pigs) in the lowlands may pose a major challenge for long-term management perspective. Previously, it was emphasized that European pigs possessed both Asian and European mtDNA resulting from the extensive history of interbreeding with predominantly Asian mtDNA introgression [3, 62, 63], which currently accounts for approximately 20–35% Asian contribution to western modern breeds [64–66]. Therefore, this concurrent maternal introgression of the global breeds, which currently accounts for about 30% in the D2 haplogroup and is widely distributed in Chinese pigs, cannot be rejected as it was evident in the maternal signatures of domestic pigs throughout most of Asia.

Our results showed a high proportion of Philippine pig haplotypes (49.07%; 53/108 individuals) belonging to D7 haplogroup compared to the previously identified similar haplotypes in MSEA (34.62%; 45/130 individuals). This haplogroup has not been previously reported in Chinese pigs, which precludes the origin of this haplogroup out of China [10]. As previously reported, this haplogroup was the most recent pig mtDNA lineage to be discovered and is distinct from those in previously documented centers of pig domestication. Because of the absence of a similar haplotype in the Insular and NEA regions, the phylogeographic origin of Phil-D7 (Philippine type D7 haplotypes) presents an interesting question because distribution of haplotypes does not primarily point to the IBBH as the direct origin of Phil-D7 or to the likelihood that these haplotypes arose through human-mediated introduction into the Philippines. Furthermore, the pattern of distribution of the haplotypes is not consistent with the hypothesized migratory route of the Neolithic Austronesian-speaking populations, nor with the possible pig ancestral diffusion that arrived in the Philippines from the ISEA sometime during the interglacial periods of the Pleistocene. While recent studies combining pig mitochondrial DNA and morphometric data have shown human-mediated dispersal to some islands of Southeast Asia [18, 28, 30, 67], there is currently no genetic data or archaeological material to support prehistoric translocation of Philippine pigs between islands in the archipelago [29, 68]. However, assuming that the haplotypes between these two geographic locations originated from a single ancestral lineage, one could hypothesize that the significant differentiation of populations, despite shared haplotypes, could be suspected as a consequence of geographical isolation. The vicariance caused by the sharp drop of sea levels during the Quaternary has resulted in the isolation of populations due to the formation of geographic barriers to migration, and consequent

genetic divergence between these populations [68]. In addition, as predicted by the theory of genetic isolation by distance, population differentiation usually occurs when there is migration of a certain population away from its founder population, leading to a reduction in genetic diversity. Moreover, an expansion model that assumes a single founder event predicts that patterns of genetic diversity in population can be well explained by their geographic expansion from the founders, which is accompanied by genetic differentiation [69].

The haplotype diversity of the Philippine domestic pig population studied was generally moderate and similar to that in MSEA countries such as Laos and Myanmar, while it was relatively higher compared to that in Cambodia, Bhutan, and Vietnam. However, nucleotide diversity was remarkably higher in Philippine domestic pigs. As emphasized earlier, nucleotide diversity is a more suitable parameter than haplotype diversity to estimate genetic diversity in a population [70] because it takes into account both the frequency of haplotypes and the nucleotide differences between haplotypes. These values were relatively higher in the previously reported nucleotide diversity of pigs in southern China including Yunnan, the Tibetan highlands, the extensive basins of the Yangtze and Yellow Rivers, Taiwan and some Pacific Islands. These are similar to those found in the outlying areas of ISEA, Korea [66, 71], and Bhutan. This pattern of genetic variation suggests a scenario that reflects past expansion dynamics from the species' area of origin [72], which may be due to later translocations by humans and the effects of introgression between different DNA lineages [73]. In addition, Scandura et al. [72] have previously discussed that the current large-scale pattern of genetic variability in *Sus* may be associated with one or more ancient long-distance colonization events followed by divergence of isolated lineages, geographical extinction due to local extinction within a previously continuous distributional range, and isolation by distance resulting in restricted gene flow.

The genetic fixation observed in the Philippine domestic pig populations studied, reported as $F_{ST}$, indicated that gene flow between population is limited, suggesting that populations are genetically isolated from each other between regions. Nevertheless, this is an expected population scenario for Philippine domestic pigs where the natural genetic exchange is limited due to the archipelagic geographical setting of the Philippines. During the past glacial periods, the Philippine archipelago was believed to have never been connected to the Asian Continent [15–18], which has influenced restricted genetic exchange and mtDNA distribution of pigs throughout the islands.

The historical demography of Philippine domestic pig populations has been studied using mismatch distributions that represent the frequency distribution of pairwise differences among all haplotypes studied. Theoretical studies have shown that population bottlenecks and population expansions have a sound effect on the pattern of genetic polymorphism among haplotypes in the population [42]. The multimodal pattern of mismatch distribution in Philippine domestic pigs suggests that the population has undergone irrelevant demographic expansion that has occurred over a long period of time, and that the population may have been stable or declining. In addition, the neutrality test based on both Tajima's $D$ ($p = 0.05$) and Fu's $F_S$ statistics ($p = 0.05$) was not consistent with the recent population and demographic expansions. The Fu's $F_S$ test is highly sensitive to demographic expansion, resulting in large negative $F_S$ values, while the significant Tajima's $D$ value could be a sign of population expansion and bottleneck [74–76]. The $R^2$ statistics and the simulation based on coalescence process, quantified by the Raggedness index confirmed the mismatch distribution of the studied populations. The haplotypic and genealogical relationships represented in the reduced-median network showed no geographic structuring, except for a few star-like patterns, despite a significant subdivision of the populations. In general, a population that has undergone a recent population expansion displays a star-like structure in a network tree, a smooth and unimodal mismatch distribution [45], since most alleles descend from one or a few ancestral types [42].

Therefore, the reflected mismatch distribution may be a signature of the presence of different haplogroups rather than demographic stability.

The Bayesian skyline plot of the Philippine domestic pigs revealed a decrease in population during the interglacial periods of the Late Pleistocene. The cyclic fluctuations of sea level during these periods are considered to be one of the most important events shaping the contemporary geographic distribution of genetic variation and evolutionary dynamics of the population [77–79]. Moreover, the alleged decline in population of some animals is due to glacial-interglacial episodes [80, 81]. This is a consistent population scenario that occurred in both Asian and European wild boars, where population bottlenecks occurred during the Last Glacial Maximum (LGM; ~20,000 years ago). A considerable decline in population size was more pronounced in Europe than in Asia, resulting in the low genetic diversity in modern European wild boars [64]. The recent ice age also caused a huge sea level dropped about 120 m below the present levels, exposing vast areas as dry land, but the Philippines remained isolated by deep channels [17]. It is conceivable that this influenced the land distribution of pigs in the Philippine archipelago, leading to their geographic isolation and subsequently restricted gene flow. The effects of bottlenecks are evident in populations occupying smaller geographic ranges which are susceptible to stochastic events and genetic drift compared to larger and more widely distributed populations [82]. However, since the Philippine samples included historically isolated populations that may have caused bias on demographic estimates [83], it could also be that the result reflected here is merely changes in the degree of structure rather than changes in population size. Our finding of a population expansion in MSEA pig was in contrast to the population scenario reported by Frantz et al. [54], and a more severe bottleneck in ISEA during the Pleistocene. These population declines are consistent with temperature decline during this period that may have reduced overall forest cover in these areas [54, 84, 85].

## Conclusion

This study provided important insights that will properly help address the contradicting hypothesis of a possible human-mediated translocation and exchange of domestic pigs in the Philippines. The results of our study may support the Neolithic-Austronesian expansion model, while a more thorough investigation should be conducted to relate the possible ancestral diffusion that occurred from MSEA to the Philippines via Sundaland. The unique geographic setting of the Philippines has resulted in an insignificant migration of pigs and as a long-term consequence, geographical isolation had occurred. The underlying population decline of the population studied as predicted in the BSP markedly followed the LGM period. Ultimately, the escalating rate of hybridization of Philippine domestic pigs with commercial stock in the Philippines poses a serious threat to local pig populations. Therefore, urgent conservation measures and suitable management of their genetic pool are crucial to the management of animal genetic resources at the local and global levels. For future perspectives, Y-specific markers could be performed to assess the extent of male-mediated introgression from European pigs into Philippine domestic pigs.

## Supporting information

**S1 Fig. Variable positions among haplotypes of the partial mitochondrial DNA control region (about 510 bp) found in this study.** Dots (.) indicates matches with the nucleotide sequence GenBank accession number AB041480 (Main cluster of Asian origin). PWP = Philippine wild pigs; D7 = previously described as MTSEA haplogroup. Nucleotide positions are numbered according to our sequence alignment.
(TIF)

**S2 Fig.** Mismatch distributions of mitochondrial DNA sequences of the (A) Philippine domestic pigs, (B) mainland SEA pigs based on pairwise nucleotide differences.
(TIF)

**S3 Fig. Mismatch distributions of mitochondrial DNA sequences of countries in mainland Southeast Asian pigs based on pairwise nucleotide differences.**
(TIF)

**S4 Fig.** Bayesian skyline plots showing effective population size of (A) Philippine and (B) mainland Southeast Asian pigs. Median estimates of female effective population size (Nef) are shown as solid thick line (blue) and the light-blue shaded area marks the 95% credibility intervals. The abscissa is scaled in thousands of years before present (BP). The Philippine pigs revealed a long stationary period of effective population size and the population decrease event occurred roughly at about ~25,000 BP.
(TIF)

**S5 Fig. Proposed route of dispersal and human-mediated translocation of pigs in the Philippines.**
(TIF)

**S1 Table. List of samples used in the study.**
(XLSX)

**S2 Table. Newly generated Philippine pig haplotypes and the publicly available sequences of pigs found in the mainland Southeast Asia.**
(XLSX)

**S3 Table. Publicly available global pig haplotype sequences used to infer phylogenetic and network haplotypes analysis.**
(XLSX)

## Acknowledgments

We are indebted to Dr. Lawrence M. Liao for technical support and numerous insights that greatly improved this work. We thank Cyrill John Prima Godinez, Jant Cres Caigoy, Sweet Charish Godinez and all the members of the Animal Genetics Laboratory of Hiroshima University. We thank the Visayas State University, Capiz State University, Aklan State University, Iloilo State College of Fisheries, farmers, and backyard pig raisers throughout the Visayas for the help during our sampling.

## Author Contributions

**Conceptualization:** John King N. Layos, Ronel B. Geromo, Dinah M. Espina, Masahide Nishibori.

**Data curation:** John King N. Layos, Ronel B. Geromo.

**Formal analysis:** John King N. Layos.

**Funding acquisition:** John King N. Layos, Masahide Nishibori.

**Investigation:** John King N. Layos.

**Methodology:** John King N. Layos, Masahide Nishibori.

**Resources:** Masahide Nishibori.

**Software:** John King N. Layos.

**Supervision:** Dinah M. Espina, Masahide Nishibori.

**Validation:** John King N. Layos.

**Writing – original draft:** John King N. Layos, Masahide Nishibori.

**Writing – review & editing:** John King N. Layos, Ronel B. Geromo, Dinah M. Espina, Masahide Nishibori.

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
