## [Decision Letter · Decision Letter 0]

6 Oct 2021

PONE-D-21-18979Insights on the historical biogeography of Philippine native pigs and its relationship with Continental domestic and wild boarsPLOS ONE

Dear Dr. Layos,

Thank you for submitting your manuscript to PLOS ONE. After careful consideration, we feel that it has merit but does not fully meet PLOS ONE’s publication criteria as it currently stands. Therefore, we invite you to submit a revised version of the manuscript that addresses the points raised during the review process.

We look forward to receiving your revised manuscript.

Kind regards,

Bi-Song Yue, Ph.D

Academic Editor

PLOS ONE

Journal Requirements:

2. In your Methods section, please provide additional location information, including geographic coordinates for the data set if available.

3. You indicated that you had ethical approval for your study. In your Methods section, please ensure you have also stated whether you obtained consent from owners of the domestic pigs included in the study or whether the research ethics committee or IRB specifically waived the need for their consent."""

4. In your Methods section, please provide additional details regarding the wild pigs used in your study and ensure you have described the source. For more information regarding PLOS' policy on materials sharing and reporting, see https://journals.plos.org/plosone/s/materials-and-software-sharing#loc-sharing-materials.

Reviewers' comments:

Reviewer's Responses to Questions

**Comments to the Author**

1. Is the manuscript technically sound, and do the data support the conclusions?

Reviewer #1: Partly

Reviewer #2: Yes

Reviewer #3: Partly

2. Has the statistical analysis been performed appropriately and rigorously? 

Reviewer #1: Yes

Reviewer #2: Yes

Reviewer #3: I Don't Know

3. Have the authors made all data underlying the findings in their manuscript fully available?

Reviewer #1: Yes

Reviewer #2: Yes

Reviewer #3: Yes

4. Is the manuscript presented in an intelligible fashion and written in standard English?

Reviewer #1: Yes

Reviewer #2: No

Reviewer #3: Yes

5. Review Comments to the Author

Reviewer #1: The subject manuscript reports on a study evaluating the origin, dispersal, and level of genetic introgression in Philippine “native” pigs using mitochondrial DNA D-loop analysis. These genetic data are then compared to comparable mtDNA data from domestic pigs and Eurasian wild boar. While the stated goal of the study is straightforward, the exact composition of the pigs being investigated is not completely clear.

There are no “native” Sus scrofa, either wild or domestic, in the Philippines. Meijaard and Melletti (2018) list the endemic (“native”) wild pigs that exist in the Philippine archipelago as including the following: Visayan warty pig (Sus cebifrons), Philippine warty pig (Sus philippensis), Mindoro warty pig (Sus oliveri), and Palawan bearded pig (Sus ahoenobarbus). Fossils of these Philippine wild pig species variously date as far back as the Pliocene through the late Pleistocene. The earliest documented archaeological remains of domestic pigs belonging to the species Sus scrofa in the Philippines were from a site in the Cagayan Valley dating back to 4000 BP (Heaney et al. 2016). Hybridization between domestic pigs (Sus scrofa) and Visayan warty pigs (Melletti et al. 2018), Philippine warty pigs (Meijaard and Melletti 2018), and Mindoro warty pigs (Tabaranza et al. 2018) in the Philippines has been documented. More specifically, such hybridization between domestic/feral Sus scrofa and Philippine warty pigs has been observed on Luzon, Minanao, Basilan and other islands in the archipelago (Heaney et al. 2005).

All of that said, it solicits the question as to what were the pigs that this study was looking at. Given the comparisons to genetic data from known breeds of domestic pigs and Eurasian wild boar, the study would seem to be looking at only the introduced domestic pigs belonging to the species Sus scrofa. To that end, the manuscript needs to not use the modifier “native” for these domestic pigs, and these pigs should be specifically identified as “domestic” (or “local domestic”, “rural domestic”, or “local rural domestic” or something like that; the modifier “native” is misleading). On Line 113, it was noted that samples were also collected from Philippine “wild pigs”. Further, ID sample numbers P24 and P25 in Table S1 are listed as wild pigs from the Philippines. Accession No. MN625830 in Table S2 is listed as a “Philippine wild pig.” What were these Philippine wild pigs, feral Sus scrofa? There are no Eurasian wild boar present in the Philippines. Were these “wild pigs” in actuality warty pigs and not feral Sus scrofa? The two pigs illustrated in Fig. 1 A and G appear to be immature pure or hybrid warty pigs and not Sus scrofa. In a number of places in the manuscript the authors refer to “pigs”. This noun should be more specifically defined (e.g., “pigs, which includes both wild and domestic pigs” or “pigs, which includes only domestic pigs”). In most of these places, the reader is left guessing as to what types of pigs are being discussed. A single explanatory statement on this can be made in the introduction, which would then clarify the subsequent use of the noun “pig.” The authors also need to address the potential impacts of genetic introgression of warty pigs through hybridization with domestic Sus scrofa in the Philippines on their results. Is this something that was even considered in the analysis of the data?

To my knowledge, this is the first study that has specifically looked at genetic composition as related to the anthropogenic origin of domestic pigs in the Philippines. As such, the data encompassed by the subject manuscript are both unique and important. Overall, the manuscript is well written. The editorial comments noted below are largely intended as suggested changes and should be used by the authors as deemed appropriate:

• Line 1 – Does “Philippine native pigs” refer to wild or domestic pigs? There are no “native” domestic pigs in the Philippine islands.

• Line 39 – Suggest changing “native” to “domestic”

• Line 45 – Does this refer to domestic pigs or both domestic and wild pigs?

• Line 51 - Does this refer to domestic pigs or both domestic and wild pigs?

• Line 77 - I assume that this is referring to wild Sus scrofa (Yes?)

• Line 78 - Suggest changing “exists” to “exist”; as a species, Sus scrofa would be a plural subject.

• Lines 78-79 - Does “S. scrofa that exists in the Philippines today” mean the wild/feral or domestic Sus scrofa that exist in the Philippines today?

• Lines 96-97 – Does “some modern and ancient Philippine pigs” refer to wild or domestic pigs or both?

• Line 99 – Suggest changing “pigs” to “domestic pigs”

• Line 103 - Suggest changing “Philippine pigs” to “Philippine domestic pigs”

• Line 113 – Referring to “Philippine native pigs”, does this mean domestic pigs? If so, then use the modifier domestic and not native. Looking in the Supplemental tables, this must mean domestic pigs.

• Line 139 – Suggest changing “Philippine native pigs” to “Philippine domestic pigs”

• Line 177 – Does “Philippine pigs” refer to wild or domestic pigs or both?

• Line 196 – Remove the extra space between “test” and “[39]”

• Line 219 – Suggest changing “Philippine native pigs” to “Philippine domestic pigs”

• Line 230 – The “native and domestic pigs” are both domestic pigs (Yes?)

• Line 238 - Suggest changing “Philippine native pigs” to “Philippine domestic pigs”

• Line 244 – Insert a comma after “0.01)” (0.01), which)

• Line 254 - Suggest changing “Philippine native pigs” to “Philippine domestic pigs”

• Line 261 - Suggest changing “Philippine native pigs” to “Philippine domestic pigs”

• Line 272 - Suggest changing “Philippine native pigs” to “Philippine domestic pigs”

• Line 272-273 – Should “Asian and European pigs” be changed to “Asian and European domestic pigs” or does that sample also include data from Eurasian wild boar?

• Lines 293-299 - Could the animal discussed in this portion of the text in reality have been a warty pig and not a wild/feral Sus scrofa? This species difference might explain these genetic variances. Where is this animal (PHL21) listed in the Supplementary tables?

• Line 301 - Suggest changing “Philippine native pigs” to “Philippine domestic pigs”

• line 302 – Suggest changing “continental pigs and wild boar” to “continental domestic pigs and wild boar”

• Line 304 – Suggest changing “Philippine pigs” to “Philippine domestic pigs” unless this includes both domestic and wild animals

• Line 306 – What does “endemic Philippine wild pigs” refer to? Feral Sus scrofa? Native/endemic warty and/or bearded pigs?

• Line 337 – Suggest changing “Philippine native pigs” to “Philippine domestic pigs”

• Line 339 - Suggest changing “Philippine native pigs” to “Philippine domestic pigs”

• Line 358 - Suggest changing “Philippine native pigs” to “Philippine domestic pigs”

• Lines 369-370 - Suggest changing “Philippine native pigs” to “Philippine domestic pigs”

• Line 382 - Suggest changing “Philippine native pigs” to “Philippine domestic pigs”

• Line 384 – The acronym “NEA” needs to be defined

• Line 385 – Suggest changing “Philippine pig genetic pool” to “Philippine domestic pig genetic pool” unless this includes both domestic and wild animals

• Line 409 - Suggest changing “Philippine pig genetic pool” to “Philippine domestic pig genetic pool”

• Line 410 - Suggest changing “Philippine native pigs” to “Philippine domestic pigs”

• Line 411 – Suggest changing “Chinese pigs” to “Chinese domestic pigs”

• Line 413 – Suggest changing “Philippine native pig” to “Philippine domestic pig”

• Lines 414-415 - Suggest changing “Philippine native pigs” to “Philippine domestic pigs”

• Line 418 – Suggest changing “domestic and native pigs” to “domestic pigs”

• Line 422 - Suggest changing “Philippine native pigs” to “Philippine domestic pigs”

• Lines 459-462 - Were the wild pigs referred to in this portion of the text referring to warty and/or bearded pigs or feral Sus scrofa?

• Line 465 – Suggest changing “Philippine native pigs” to “Philippine domestic pigs”

• Line 467 - Suggest changing “Philippine native pig” to “Philippine domestic pig”

• Line 470 - Suggest changing “Philippine native pigs” to “Philippine domestic pigs”

• Line 485 – Suggest changing “Philippine native pig” to “Philippine domestic pig”

• Line 488 - Suggest changing “Philippine native pigs” to “Philippine domestic pigs”

• Line 496 – Suggest changing “Philippine native pig” to “Philippine domestic pig”

• Line 501 - Suggest changing “Philippine native pigs” to “Philippine domestic pigs”

• line 547 - Suggest changing “Philippine native pigs” to “Philippine domestic pigs”

• Line 593 – Change “2017” to “2018”; the copyright date of this book is 2018. Look at the page after the title page in the book.

• Line 607 – “Sus” needs to be italicized

• Line 630 - “Sus” needs to be italicized

• Line 699 - “Sus scrofa” needs to be italicized

• Line 712 - “Sus” needs to be italicized

• Lines 764-765 - “Sus scrofa” needs to be italicized

References:

Heaney, L. R., D. S. Balete, G. V. Gee, et al. 2005. Preliminary report on the mammals of Balbasang, Kaling Province, Luzon. Sylvatrop: The Philippine Forest Research Journal 13:51-62.

Heaney, L. R., D. S. Balete and E. A. Rickart. 2016. The mammals of Luzon Island: biogeography and natural history of a Philippine fauna. Johns Hopkins University Press, Baltimore, Maryland, USA. 304 pp.

Melletti, M., and E. Meijaard (editors). 2018. Ecology, Evolution and Management of Wild Pigs and Peccaries: Implications for Conservation. Cambridge University Press, United Kingdom. 480 pp.

Meijaard, E., and M. Melletti. 2018. Philippine warty pig Sus philippensis (Nehring, 1886). pp. 157-161. in Ecology, Evolution and Management of Wild Pigs and Peccaries: Implications for Conservation. (M. Melletti and E. Meijaard, eds.). Cambridge University Press, United Kingdom. 480 pp.

Melletti, M., E. Meijaard, and L. Przybylska. 2018. Visayan warty pig Sus cebifrons (Heude, 1888). pp. 150-156. in Ecology, Evolution and Management of Wild Pigs and Peccaries: Implications for Conservation. (M. Melletti and E. Meijaard, eds.). Cambridge University Press, United Kingdom. 480 pp.

Tabaranza, D. G. E., E. Schütz, J. C. T. Gonzalez, L. M. Espiritu-Afuang. 2018. Mindoro warty pig Sus oliveri (Groves, 1997). pp. 162-169. in Ecology, Evolution and Management of Wild Pigs and Peccaries: Implications for Conservation. (M. Melletti and E. Meijaard, eds.). Cambridge University Press, United Kingdom. 480 pp.

Reviewer #2: The authors produce some interesting results that should be of interest to PLOS ONE readership. It is original research that has been ethically conducted with robust statistical methods. They contributed their genetic samples to GenBank and provided all of their supplemental materials for download. I believe if someone was so inclined they would be able to reproduce their research which is the hallmark of sound science.

The only thing that sticks out to me is that English is not their first language. There are no outright errors, it just doesn't flow well. If they could find a native English speaker to polish it a bit then it would be fantastic. My recommendation is a minor revision.

Reviewer #3: This manuscript is an important contribution to the discussion of human and human-mediated plant/animal global dispersion. I feel that the manuscript could benefit from some re-organization, refinement, and a bit more detail in the introduction.

- Line 48 - What is the D7 haplogroup? I’m not sure the authors can refer to this without describing what it is.

- LINE 50 - What is the significance of a D2 haplogroup how would the reader know what these are?

- Line 54-56 - It seems like a sentence may be missing or the phrasing is off. Perhaps another sentence concluding the significance of the work is needed.

- Introduction:

- The authors should consider breaking the introduction up into more paragraphs to help the reader.

- The authors should also consider adding in some discussion of the D-type haplogroups as it plays a very important role later on in the manuscript, but is not described at all until the Discussion.

- For example, lines 370-378 may be more appropriate for the introduction so the reader has some sense as to why, in the results section, there is reference to “D-type” haplotypes.

- Furthermore, which D cluster refers to Asian Phologeographic vs. European?? (See line 380)

- Line 222 - S4 appendix does not appear in the manuscript that the review has access to.

- Line 276 - It is not clear to me in Fig 2 the two distinct core lineages. Suggest that the authors revise Fig 2 to clearly indicate the “asian” and “European” core lineages.

- Line 281 - The authors make it very difficult to follow the groupings by creating multiple different short-hand methods to describe geographic groupings. In line 281, the authors refer to MTSEA as D7. Suggest sticking to one of these classifications.

- S2 Fig - It would be helpful to have the countries labeled in this map so that the reader can refer to their specific locations when reviewing Table 1 as well as the introduction, specifically lines 78-86

- Line 384 - What is “NEA”? I don’t see it defined anywhere else. The number of acronyms and confusing usage of them has mad reading this manuscript very challenging. Strongly suggest removing all acronyms or simplifying significantly.

- Line 397 - Perhaps the authors should consider introducing the importance of the “D” haplotypes to their results in the introduction so that the reader may understand the classification

- Line 393-398 - Seems like a key finding of the study, but it is buried in the middle of a large paragraph.

- Perhaps the map could also include the geographic origins of the D-type haplotype clusters. The phylogenetic tree might also

- I’m having some trouble understanding how the philippine “native” pigs that were presumably dispersed by humans (whether from Taiwan or otherwise) can be considered “native” and would want the authors to define what they consider to be native and at what point of genetic mixing would the animal no longer be classified as native? Further, in line 459 the authors refer to “wild” Philippine pigs. Are these different than “native” pigs?

- The authors might consider bringing an abridged summary of lines 443-458 into the introduction to further help the reader understand the context of this manuscript.

6. PLOS authors have the option to publish the peer review history of their article (what does this mean?). If published, this will include your full peer review and any attached files.

Reviewer #1: No

Reviewer #2: No

Reviewer #3: No

---

## [Author Response · Author response to Decision Letter 0]

29 Oct 2021

Reviewer #1: The subject manuscript reports on a study evaluating the origin, dispersal, and level of genetic introgression in Philippine “native” pigs using mitochondrial DNA D-loop analysis. These genetic data are then compared to comparable mtDNA data from domestic pigs and Eurasian wild boar. While the stated goal of the study is straightforward, the exact composition of the pigs being investigated is not completely clear.

There are no “native” Sus scrofa, either wild or domestic, in the Philippines. Meijaard and Melletti (2018) list the endemic (“native”) wild pigs that exist in the Philippine archipelago as including the following: Visayan warty pig (Sus cebifrons), Philippine warty pig (Sus philippensis), Mindoro warty pig (Sus oliveri), and Palawan bearded pig (Sus ahoenobarbus). Fossils of these Philippine wild pig species variously date as far back as the Pliocene through the late Pleistocene. The earliest documented archaeological remains of domestic pigs belonging to the species Sus scrofa in the Philippines were from a site in the Cagayan Valley dating back to 4000 BP (Heaney et al. 2016). Hybridization between domestic pigs (Sus scrofa) and Visayan warty pigs (Melletti et al. 2018), Philippine warty pigs (Meijaard and Melletti 2018), and Mindoro warty pigs (Tabaranza et al. 2018) in the Philippines has been documented. More specifically, such hybridization between domestic/feral Sus scrofa and Philippine warty pigs has been observed on Luzon, Mindanao, Basilan and other islands in the archipelago (Heaney et al. 2005).

We have added information in the introduction about the numerous endemic Philippine wild pigs, as well as the hypothesized introduction of domestic pig in the Philippines. 

The following citations were also added:

1. Oliver WLR. The taxonomy, distribution and status of Philippine wild pigs. Ibex, Journal of Mountain Ecology. 1995;3: 26–32. 

2. Luskin MS, Ke A. Bearded pig Sus barbatus (Müller,1838). In: Meletti M, Meijaard E, editors. Ecology, Conservation and Management of Wild Pigs and Peccaries. Cambridge University Press; 2018. pp. 176. 

3. Piper PJ, Hung HC, Campos FZ, Bellwood P, Santiago R. A 4000-year-old introduction of domestic pigs into the Philippine archipelago: Implications for understanding routes of human migration into through Island Southeast Asia and Wallacea. Antiquity. 2009;83: 687–695. 

4. Amano N, Piper PJ, Hung HC, Bellwood P. Introduced domestic animals in the Neolithic and Metal Age of the Philippines: Evidence from Nagsabaran, Northern Luzon. J. Island Coast. Archaeol. 2013. 8, 317-335.

All of that said, it solicits the question as to what were the pigs that this study was looking at. Given the comparisons to genetic data from known breeds of domestic pigs and Eurasian wild boar, the study would seem to be looking at only the introduced domestic pigs belonging to the species Sus scrofa. To that end, the manuscript needs to not use the modifier “native” for these domestic pigs, and these pigs should be specifically identified as “domestic” (or “local domestic”, “rural domestic”, or “local rural domestic” or something like that; the modifier “native” is misleading). On Line 113, it was noted that samples were also collected from Philippine “wild pigs”. Further, ID sample numbers P24 and P25 in Table S1 are listed as wild pigs from the Philippines. Accession No. MN625830 in Table S2 is listed as a “Philippine wild pig.” What were these Philippine wild pigs, feral Sus scrofa? There are no Eurasian wild boar present in the Philippines. Were these “wild pigs” in actuality warty pigs and not feral Sus scrofa? The two pigs illustrated in Fig. 1 A and G appear to be immature pure or hybrid warty pigs and not Sus scrofa. In a number of places in the manuscript the authors refer to “pigs”. This noun should be more specifically defined (e.g., “pigs, which includes both wild and domestic pigs” or “pigs, which includes only domestic pigs”). In most of these places, the reader is left guessing as to what types of pigs are being discussed. A single explanatory statement on this can be made in the introduction, which would then clarify the subsequent use of the noun “pig.” The authors also need to address the potential impacts of genetic introgression of warty pigs through hybridization with domestic Sus scrofa in the Philippines on their results. Is this something that was even considered in the analysis of the data?

We would like to apologize for the confusion in the modifiers used in the manuscript. We have changed all modifiers "native pig" to "domestic pig" throughout the manuscript where appropriate to avoid further confusion. The Philippine wild pig described in this study is indeed a wild pig. In this case, we did not use the term "warty pigs" in the discussion because the classification of the wild pig analyzed is not yet known. Therefore, to be on the safe side, we use the term "wild pigs" throughout the dataset. Figures 1A and B are Philippine domestic pigs, not warty pigs. As a future direction for this research, we intend to analyze the potential impact of wild pigs on domestic pigs in the Philippines, as we did not conduct such an analysis in this study.

To my knowledge, this is the first study that has specifically looked at genetic composition as related to the anthropogenic origin of domestic pigs in the Philippines. As such, the data encompassed by the subject manuscript are both unique and important. Overall, the manuscript is well written. The editorial comments noted below are largely intended as suggested changes and should be used by the authors as deemed appropriate:

We appreciate your positive feedback on our study. Yes, this study is the first screening of mtDNA D-loop variation from Philippine domestic pigs to clarify their distribution and past dispersal history. In the future, the authors would like to conduct further relevant research in this field to bridge the gap between the highly controversial anthropogenetic causes of diversity in the Philippines, especially in domestic pigs. We believe that this publication is a start so that we can attract future collaborators for a larger research project.

Line 1 – Does “Philippine native pigs” refer to wild or domestic pigs? There are no “native” domestic pigs in the Philippine islands.

Revised accordingly. We changed the modifier “native pigs” to “domestic pigs”.

Line 39 – Suggest changing “native” to “domestic”

Revised accordingly (see Line 46 in the revised manuscript).

Line 45 – Does this refer to domestic pigs or both domestic and wild pigs?

We are referring to Philippine domestic pigs. Changed to Philippine domestic pigs as suggested.

Line 51 - Does this refer to domestic pigs or both domestic and wild pigs?

The authors are referring to Philippine domestic pigs. Changed to Philippine domestic pigs as suggested.

Line 77 - I assume that this is referring to wild Sus scrofa (Yes?)

Yes.

Line 78 - Suggest changing “exists” to “exist”; as a species, Sus scrofa would be a plural subject.

Revised accordingly.

Lines 78-79 - Does “S. scrofa that exists in the Philippines today” mean the wild/feral or domestic Sus scrofa that exist in the Philippines today?

This phrase has been changed to “Therefore, the ubiquitous wild boar S. scrofa was not considered native to the Philippines because it was unable to reach the archipelago [15-19] and was probably introduced as a domestic animal within the last few thousand years [19]” (see Lines 94-96 in the revised manuscript).

Lines 96-97 – Does “some modern and ancient Philippine pigs” refer to wild or domestic pigs or both?

The authors are referring to Philippine domestic pigs (see Lines 116-118 in the revised manuscript).

Line 99 – Suggest changing “pigs” to “domestic pigs”

Revised accordingly (see Line 119 in the revised manuscript).

Line 103 - Suggest changing “Philippine pigs” to “Philippine domestic pigs”

Revised accordingly (see Line 132 in the revised manuscript).

Line 113 – Referring to “Philippine native pigs”, does this mean domestic pigs? If so, then use the modifier domestic and not native. Looking in the Supplemental tables, this must mean domestic pigs.

Revised accordingly (see Line 143 in the revised manuscript).

Line 139 – Suggest changing “Philippine native pigs” to “Philippine domestic pigs”

Revised accordingly (see Line 177 in the revised manuscript).

Line 177 – Does “Philippine pigs” refer to wild or domestic pigs or both?

Both. (see Line 216 in the revised manuscript)

Line 196 – Remove the extra space between “test” and “[39]”

Revised accordingly (see Line 235 in the revised manuscript).

Line 219 – Suggest changing “Philippine native pigs” to “Philippine domestic pigs”

Revised accordingly (see Line 258 in the revised manuscript).

Line 230 – The “native and domestic pigs” are both domestic pigs (Yes?)

Yes. “native and” was deleted (see Line 269 in the revised manuscript).

Line 238 - Suggest changing “Philippine native pigs” to “Philippine domestic pigs”

Revised accordingly (see Line 277 in the revised manuscript).

Line 244 – Insert a comma after “0.01)” (0.01), which)

Revised accordingly (see Line 283 in the revised manuscript).

Line 254 - Suggest changing “Philippine native pigs” to “Philippine domestic pigs”

Revised accordingly (see Line 293 in the revised manuscript).

Line 261 - Suggest changing “Philippine native pigs” to “Philippine domestic pigs”

Revised accordingly (see Line 300 in the revised manuscript).

Line 272 - Suggest changing “Philippine native pigs” to “Philippine domestic pigs”

Revised accordingly (see Line 311 in the revised manuscript).

Line 272-273 – Should “Asian and European pigs” be changed to “Asian and European domestic pigs” or does that sample also include data from Eurasian wild boar?

This samples include Eurasian wild boars (see Lines 311-312 in the revised manuscript).

Lines 293-299 - Could the animal discussed in this portion of the text in reality have been a warty pig and not a wild/feral Sus scrofa? This species difference might explain these genetic variances. Where is this animal (PHL21) listed in the Supplementary tables?

This animal is actually a warty pig and not a feral/wild Sus scrofa. However, in this case we used the general term "Philippine wild pigs" for this haplotype because its species classification has not yet been identified (see Line 331 in the revised manuscript). This haplotype/sample can be found in S1 Table ID #P24&25, S2 Table #67 under accession number MN625830.

Line 301 - Suggest changing “Philippine native pigs” to “Philippine domestic pigs”

Revised accordingly (see Line 339 in the revised manuscript).

Line 302 – Suggest changing “continental pigs and wild boar” to “continental domestic pigs and wild boar”

Revised accordingly (see Line 340 in the revised manuscript).

Line 304 – Suggest changing “Philippine pigs” to “Philippine domestic pigs” unless this includes both domestic and wild animals.

Revised accordingly (see Line 342 in the revised manuscript).

Line 306 – What does “endemic Philippine wild pigs” refer to? Feral Sus scrofa? Native/endemic warty and/or bearded pigs?

The authors are referring to endemic Philippine wild pigs (see Line 343 in the revised manuscript).

Line 337 – Suggest changing “Philippine native pigs” to “Philippine domestic pigs”

Revised accordingly (see Lines 374-375 in the revised manuscript).

Line 339 - Suggest changing “Philippine native pigs” to “Philippine domestic pigs”

Revised accordingly (see Line 376 in the revised manuscript).

Line 358 - Suggest changing “Philippine native pigs” to “Philippine domestic pigs”

Revised accordingly (see Line 396 in the revised manuscript).

Lines 369-370 - Suggest changing “Philippine native pigs” to “Philippine domestic pigs”

Revised accordingly (see Lines 407-408 in the revised manuscript). 

Line 382 - Suggest changing “Philippine native pigs” to “Philippine domestic pigs”

Revised accordingly (see Line 411 in the revised manuscript).

Line 384 – The acronym “NEA” needs to be defined.

Revised accordingly (see Line 413 in the revised manuscript).

Line 385 – Suggest changing “Philippine pig genetic pool” to “Philippine domestic pig genetic pool” unless this includes both domestic and wild animals.

Revised accordingly (see Line 414 in the revised manuscript).

Line 409 - Suggest changing “Philippine pig genetic pool” to “Philippine domestic pig genetic pool”

Revised accordingly (see Line 440 in the revised manuscript).

Line 410 - Suggest changing “Philippine native pigs” to “Philippine domestic pigs”

Revised accordingly (see Line 441 in the revised manuscript).

Line 411 – Suggest changing “Chinese pigs” to “Chinese domestic pigs”

Revised accordingly (see Line 442 in the revised manuscript).

Line 413 – Suggest changing “Philippine native pig” to “Philippine domestic pig”

Revised accordingly (see Line 444 in the revised manuscript).

Lines 414-415 - Suggest changing “Philippine native pigs” to “Philippine domestic pigs”

Revised accordingly (see Lines 445-446 in the revised manuscript).

Line 418 – Suggest changing “domestic and native pigs” to “domestic pigs”

Revised accordingly (see Lines 447 in the revised manuscript).

Line 422 - Suggest changing “Philippine native pigs” to “Philippine domestic pigs”

Revised accordingly (see Line 453 in the revised manuscript).

Lines 459-462 - Were the wild pigs referred to in this portion of the text referring to warty and/or bearded pigs or feral Sus scrofa?

The authors are referring to Philippine wild pigs (see Line 490 in the revised manuscript). All endemic wild pigs in the Philippines are commonly referred to as Philippine wild pigs unless genetically characterized.

Line 465 – Suggest changing “Philippine native pigs” to “Philippine domestic pigs”

Revised accordingly (see Line 496 in the revised manuscript).

Line 467 - Suggest changing “Philippine native pig” to “Philippine domestic pig”

Revised accordingly (see Line 498 in the revised manuscript).

Line 470 - Suggest changing “Philippine native pigs” to “Philippine domestic pigs”

Revised accordingly (see Line 501 in the revised manuscript).

Line 485 – Suggest changing “Philippine native pig” to “Philippine domestic pig”

Revised accordingly (see Line 516 in the revised manuscript).

Line 488 - Suggest changing “Philippine native pigs” to “Philippine domestic pigs”

Revised accordingly (see Line 519 in the revised manuscript).

Line 496 – Suggest changing “Philippine native pig” to “Philippine domestic pig”

Revised accordingly (see Line 524 in the revised manuscript).

Line 501 - Suggest changing “Philippine native pigs” to “Philippine domestic pigs”

Revised accordingly (see Line 529 in the revised manuscript).

Line 547 - Suggest changing “Philippine native pigs” to “Philippine domestic pigs”

Revised accordingly (see Line 573 in the revised manuscript).

Line 593 – Change “2017” to “2018”; the copyright date of this book is 2018. Look at the page after the title page in the book.

Revised accordingly (see Line 617 in the revised manuscript).

Line 607 – “Sus” needs to be italicized

Revised accordingly (see Line 637 in the revised manuscript).

Line 630 - “Sus” needs to be italicized

Revised accordingly (see Line 662 in the revised manuscript).

Line 699 - “Sus scrofa” needs to be italicized

Revised accordingly (see Line 640 in the revised manuscript).

Line 712 - “Sus” needs to be italicized

Revised accordingly (see Line 731 in the revised manuscript).

Lines 764-765 - “Sus scrofa” needs to be italicized

Revised accordingly (see Lines 778-779 in the revised manuscript).

References:

Heaney, L. R., D. S. Balete, G. V. Gee, et al. 2005. Preliminary report on the mammals of Balbasang, Kaling Province, Luzon. Sylvatrop: The Philippine Forest Research Journal 13:51-62.

Heaney, L. R., D. S. Balete and E. A. Rickart. 2016. The mammals of Luzon Island: biogeography and natural history of a Philippine fauna. Johns Hopkins University Press, Baltimore, Maryland, USA. 304 pp.

Melletti, M., and E. Meijaard (editors). 2018. Ecology, Evolution and Management of Wild Pigs and Peccaries: Implications for Conservation. Cambridge University Press, United Kingdom. 480 pp.

Meijaard, E., and M. Melletti. 2018. Philippine warty pig Sus philippensis (Nehring, 1886). pp. 157-161. in Ecology, Evolution and Management of Wild Pigs and Peccaries: Implications for Conservation. (M. Melletti and E. Meijaard, eds.). Cambridge University Press, United Kingdom. 480 pp.

Melletti, M., E. Meijaard, and L. Przybylska. 2018. Visayan warty pig Sus cebifrons (Heude, 1888). pp. 150-156. in Ecology, Evolution and Management of Wild Pigs and Peccaries: Implications for Conservation. (M. Melletti and E. Meijaard, eds.). Cambridge University Press, United Kingdom. 480 pp.

Tabaranza, D. G. E., E. Schütz, J. C. T. Gonzalez, L. M. Espiritu-Afuang. 2018. Mindoro warty pig Sus oliveri (Groves, 1997). pp. 162-169. in Ecology, Evolution and Management of Wild Pigs and Peccaries: Implications for Conservation. (M. Melletti and E. Meijaard, eds.). Cambridge University Press, United Kingdom. 480 pp.

Reviewer #2: The authors produce some interesting results that should be of interest to PLOS ONE readership. It is original research that has been ethically conducted with robust statistical methods. They contributed their genetic samples to GenBank and provided all of their supplemental materials for download. I believe if someone was so inclined they would be able to reproduce their research which is the hallmark of sound science.

The only thing that sticks out to me is that English is not their first language. There are no outright errors, it just doesn't flow well. If they could find a native English speaker to polish it a bit then it would be fantastic. My recommendation is a minor revision.

Thank you for your positive feedback on our paper and your recommendation for a minor revision. We have resubmitted our paper for English critique and proofreading to improve the manuscript. As you could notice, there were changes throughout the manuscript. 

Reviewer #3: This manuscript is an important contribution to the discussion of human and human-mediated plant/animal global dispersion. I feel that the manuscript could benefit from some re-organization, refinement, and a bit more detail in the introduction.

Line 48 - What is the D7 haplogroup? I’m not sure the authors can refer to this without describing what it is.

Revised accordingly. We added some information about the D7 haplogroup in the Abstract (see Lines 54-55 in the revised manuscript). 

LINE 50 - What is the significance of a D2 haplogroup how would the reader know what these are?

We have added quotation of D2 haplotypes in the Abstract "Asian major" (see Line 58 in the revised manuscript). Due to the limited number of words in the abstract, see Lines 323-325 and 458-461 of the revised manuscript for more details on this haplogroup. The phrase “which postulate the legitimate dispersal of pigs associated with the multiple waves of human migrations involving the Philippines” was deleted and was replaced by phrase “on several Philippine islands”. 

Line 54-56 - It seems like a sentence may be missing or the phrasing is off. Perhaps another sentence concluding the significance of the work is needed.

Revised accordingly. We have also added the concluding remarks in the abstract: " This finding highlights an important challenge in managing Philippine domestic pig genetic resources from a local and global perspective" (see Lines 62-64 in the revised manuscript).

Introduction:

- The authors should consider breaking the introduction up into more paragraphs to help the reader.

Revised accordingly. We have revised almost the entire introduction (see Lines 67-136 in the revised manuscript).

-The authors should also consider adding in some discussion of the D-type haplogroups as it plays a very important role later on in the manuscript but is not described at all until the Discussion.

Revised accordingly. We have added discussions of the various haplogroups in the introduction (see Lines 78-85 in the revised manuscript).

- For example, lines 370-378 may be more appropriate for the introduction so the reader has some sense as to why, in the results section, there is reference to “D-type” haplotypes.

Revised accordingly (see Lines 78-85 in the revised manuscript). 

- Furthermore, which D cluster refers to Asian Phylogeographic vs. European?? (See line 380)

By and large, the Sus scrofa was historically domesticated from two ancestral animals: the Asian and the European Sus scrofa wild boar. Later, the lineages of these wild boars separated, resulting in an independent evolutionary lineage known as "haplogroups." Today, the wild boars of Asian origin that are widely recognized are the D2, D3, D5, D6, D7 and the Lanyu pigs (Asian phylogeographic origin), while the European wild boars are the D1 and D4 (European phylogeographic origin) (see Fig 3).

- Line 222 - S4 appendix does not appear in the manuscript that the review has access to.

We apologize. S4 appendix refers to S1 Fig. 

- Line 276 - It is not clear to me in Fig 2 the two distinct core lineages. Suggest that the authors revise Fig 2 to clearly indicate the “Asian” and “European” core lineages.

We revised accordingly. We have also decided to re-analyze this figure. We have changed our methodology in constructing the phylogenetic tree from Maximum Likelihood to Bayesian Inference using program Mr Bayes. Please see Fig 3 in the revised manuscript.

- Line 281 - The authors make it very difficult to follow the groupings by creating multiple different short-hand methods to describe geographic groupings. In line 281, the authors refer to MTSEA as D7. Suggest sticking to one of these classifications.

Revised accordingly (see Line 320 in the revised manuscript).

- S2 Fig - It would be helpful to have the countries labelled in this map so that the reader can refer to their specific locations when reviewing Table 1 as well as the introduction, specifically lines 78-86

Revised accordingly. 

- Line 384 - What is “NEA”? I don’t see it defined anywhere else. The number of acronyms and confusing usage of them has mad reading this manuscript very challenging. Strongly suggest removing all acronyms or simplifying significantly.

Revised accordingly (see Line 413 in the revised manuscript). 

- Line 397 - Perhaps the authors should consider introducing the importance of the “D” haplotypes to their results in the introduction so that the reader may understand the classification.

Revised accordingly (see Lines 78-85 in the revised manuscript).

- Line 393-398 - Seems like a key finding of the study, but it is buried in the middle of a large paragraph.

Revised accordingly. We have decided to separate this paragraph for emphasis. We have also added citation “The complexity surrounding the Neolithic Austronesian expansion and dispersal has recently raised questions due to the limited studies conducted in the Philippine archipelago, as the faunal assemblages in this key region have provided some clues [4]. For instance,” (see Lines 426-436 in the revised manuscript).

1. Larson G, Dobney K, Albarella U, Fang M, Smith EM, Robins J, et al. Worldwide phylogeography of wild boar reveals multiple centers of pig domestication. Science. 2005;307: 1618–1621.

- Perhaps the map could also include the geographic origins of the D-type haplotype clusters. The phylogenetic tree might also.

Revised accordingly. We have added figure on this. Please see Figure 1 in the revised manuscript.

- I’m having some trouble understanding how the Philippine “native” pigs that were presumably dispersed by humans (whether from Taiwan or otherwise) can be considered “native” and would want the authors to define what they consider to be native and at what point of genetic mixing would the animal no longer be classified as native? Further, in line 459 the authors refer to “wild” Philippine pigs. Are these different than “native” pigs?

We apologize for the confusion in the modifiers used. To avoid it, we have replaced all modifiers "native" with "domestic" throughout the manuscript where appropriate. Indeed, line 459 refers to the Philippine wild pig (endemic to the Philippines), whereas "native/domestic pigs" in this study refers to the introduced pig (see Line 490 in the revised manuscript).

- The authors might consider bringing an abridged summary of lines 443-458 into the introduction to further help the reader understand the context of this manuscript. 

Revised accordingly. We have added additional citations in the introduction (see Lines 94-96, 101-103).

---

## [Decision Letter · Decision Letter 1]

17 Jan 2022

PONE-D-21-18979R1Insights on the historical biogeography of Philippine domestic pigs and its relationship with continental domestic pigs and wild boarsPLOS ONE

Dear Dr. Layos,

Thank you for submitting your manuscript to PLOS ONE and for your patience as we worked to find reviewers who could provide a meaningful evaluation of your work. After careful consideration, we feel that your revised submission would be of interest to the PLOS ONE community but it does not fully meet PLOS ONE’s publication criteria as it currently stands. Therefore, we invite you to submit a revised version of the manuscript that addresses the points raised during the review process. Both reviewers 1 and 4 feel your study helps to expand understanding of the origin and genetic composition of domesticated pigs in the Philippines. In your revision, please address the suggestions and comments both reviewers have provided to help improve the clarity and impact of the manuscript. Please also address reviewer 4's questions regarding similarity between this manuscript and another currently in review in another journal. As per PLOS ONE's publication policy, articles will not be accepted for publication if they have already been published "in whole or par, elsewhere in the peer-reviewed literature". And, all figures should be original and not published elsewhere. Please make sure to keep these policies in mind when responding to reviewer 4 and drafting your revision.

We look forward to receiving your revised manuscript.

Kind regards,

Jeffrey A. Eble, Ph.D.

Academic Editor

PLOS ONE

Journal Requirements:

Reviewers' comments:

Reviewer's Responses to Questions

**Comments to the Author**

1. If the authors have adequately addressed your comments raised in a previous round of review and you feel that this manuscript is now acceptable for publication, you may indicate that here to bypass the “Comments to the Author” section, enter your conflict of interest statement in the “Confidential to Editor” section, and submit your "Accept" recommendation.

Reviewer #1: (No Response)

Reviewer #4: (No Response)

2. Is the manuscript technically sound, and do the data support the conclusions?

Reviewer #1: Yes

Reviewer #4: Yes

3. Has the statistical analysis been performed appropriately and rigorously? 

Reviewer #1: Yes

Reviewer #4: Yes

4. Have the authors made all data underlying the findings in their manuscript fully available?

Reviewer #1: Yes

Reviewer #4: Yes

5. Is the manuscript presented in an intelligible fashion and written in standard English?

Reviewer #1: Yes

Reviewer #4: Yes

6. Review Comments to the Author

Reviewer #1: The subject revised manuscript reports on a study evaluating the origin, dispersal, and level of genetic introgression in Philippine domestic pigs using mitochondrial DNA D-loop analysis. These genetic data are then compared to comparable mtDNA data from domestic pigs and Eurasian wild boar. This revision is much improved over the earlier draft. Again, to my knowledge, this is the first study that has specifically looked at genetic composition as related to the anthropogenic origin of domestic pigs in the Philippines. As such, the data encompassed by the subject manuscript are both unique and important. Overall, the manuscript is well written. The editorial comments noted below are largely intended as suggested changes and should be used by the authors as deemed appropriate:

• Lines 43-44 – Suggest changing “the history of pig dispersal” to “the history of domestic pig (Sus scrofa) dispersal”.

• Line 45 – Suggest changing “associated with some human migration events” to “associated with human migration events”

• Line 47 – Suggest changing “wild boars” to “wild boar”

• Line 55 – Suggest changing “Philippine pig haplotypes” to “Philippine domestic pig haplotypes”

• Line 67 – Suggest changing “The wild boar (Sus scrofa L.)” to “The Eurasian wild boar (Sus scrofa L.)”

• Lines 86-87 – Suggest changing “an exceptionally high levels of endemism” to “an exceptionally high level of endemism”

• Line 87 – Suggest changing “The Philippines has repeatedly” to “The Philippines have repeatedly”

• Line 89 - Suggest changing “four endemic wild pigs such as the Philippine” to “four endemic wild pigs including the Philippine”

• Line 91 – Suggest changing “as well as one native shared with Sundaic” to “as well as one native species shared with the Sundaic”

• Line 94 – Suggest changing “ubiquitous wild boar S. scrofa was not considered native” to “ubiquitous Eurasian wild boar S. scrofa is not native”

• Lines 111-112 – Suggest changing “For this reason, [27] excluded” to “For this reason, Larson et al. [27] excluded”

• Line 128 – Suggest changing “an introduced pig breed” to “an introduced domestic pig breed”

• Line 133 - Suggest changing “diversity and structure exhibits a signature” to “diversity and structure exhibit a signature”

• Line 217 - Suggest changing “described by [4] with six clades” to “described by Larson et al. [4] with six clades”

• Line 218 – Suggest changing “proposed by [10] were used” to “proposed by Tanaka et al. [10] were used”

• Line 287 – Suggest changing “most pigs in the populations studies” to “most pigs in the population studies”

• Line 320 - Suggest changing “D7 haplogroup which accounted” to “D7 haplogroup, which accounted”

• Line 324 – Suggest changing “what [5] recognized” to “what Scandura et al. [5] recognized”

• Lines 333-334 – Suggest changing “the previous researches.” to “the previous research.”

• Line 340 – Suggest changing “wild boars” to “wild boar”

• Line 361 - Suggest changing “contrast, domestic pigs in MSEA combined revealed” to “contrast, combined domestic pigs in MSEA revealed”

• Line 404 - Suggest changing “Philippine pigs” to “Philippine domestic pigs”

• Line 437 - Suggest changing “Before the European arrived in the Philippines” to “Before the Europeans arrived in the Philippines”

• Line 442 – Suggest changing “which exhibited” to “which is exhibited”

• Line 446 - Suggest changing “was minimal, which accounted for only” to “was minimal, accounting for only”

• Line 464 - Suggest changing “Chinese pigs, therefore this precluded the origin of these haplogroup” to “Chinese pigs, which precludes the origin of this haplogroup”

• Line 480 – Suggest changing “populations despite shared haplotypes could” to “populations, despite shared haplotypes, could”

• Lines 490-497 – What “Philippine wild pigs” are being referred to here? The Philippine wild pigs that Bill Oliver was referring to in the IBEX article cited are not Sus scrofa, and in fact are all separate species from either domestic or wild Sus scrofa. Without identifying the species of the Philippine wild pigs that you are referring to, I’m not sure of the value of this paragraph to the rest of the study.

• Lines 510-511 - Suggest changing “In addition, [72] have previously discussed” to “In addition, Scandura et al. [72] have previously discussed”

• Line 525 – Suggest changing “distributions which represent” to “distributions that represent”

• Line 560 - Suggest changing “reported by [54]” to “reported by Frantz et al. [54]”

• Line 566 – Suggest changing “exchange of pigs” to “exchange of domestic pigs”

• Line 574 – What is the name “native pig populations” referring to here, domestic or wild pigs? There are no “native” domestic pig populations in the Philippines.

John J. Mayer, Ph.D.

Savannah River National Laboratory

Battelle Savannah River Alliance

Aiken, SC 29808

(803) 819-8404

john.mayer@srnl.doe.gov

Reviewer #4: This manuscript aimed to present the origin and distribution of Philippine domestic pigs based on partial mitochondrial D-loop sequences analysis. The results showed the genetic connection between the Philippine domestic and continental pigs, supporting the Neolithic-Austronesian expansion model and proposing the possible alternative migration route. The data and analyses presented in the manuscript are appropriate, and the results support the key messages. The sequences under accessions MN625805-MN625830 are searchable; however, the sequences under accessions MW924902-MW92973 could not be found in the GenBank database.

I am reviewing another manuscript submitted to a different journal. In that manuscript, some pictures of pigs looked the same as all sub-figures in Fig. 2, except Fig. 2F. In addition, some contents in the introduction section of that manuscript were very similar, if not the same, as some parts of the introduction of this manuscript; for example, the content on Page 4, Line 96-103 starts with the words "To date." However, more Philippine pig samples were included, and some analyses applied on mitochondrial D-loop sequences in that manuscript differed from this manuscript. In addition, the conclusion in that manuscript could be considered as an answer of the future direction of this manuscript mentioned on Page 23, Line 567-569, that " a more thorough investigation should be conducted to relate the possible ancestral diffusion that occurred from MSEA to the Philippines via Sundaland.

Despite the concerns in the previous paragraph, the following comments should be addressed before accepting publication.

Page 2, Line 61-62: Regarding the sentence on these lines, what is the population showing sudden decline? The population name, such as Philippine pigs, Mainland Southeast Asian pigs, should be specified. In my opinion, both Bayesian skyline plots (Fig. 7) did not clearly show a sudden decline, so I am wondering what result supported this statement? This comment also applied to the statement "The sudden population decline occurred approximately 25,000 years before present (BP)" on Page 15, Line 377-378 and "The underlying sudden population decline as predicted in the BSP markedly followed the LGM period" on Page 23, Line 571-572.

Page 8-9, Line 214-216: The statement "This method calculates the net divergence of each taxon from all other taxa as the sum of the individual distances from variance within and among groups of Philippine pigs and comparison sequences." should be revised. It was not whether the net divergence of each taxon was the sum of the individual distance, and the distance was calculated from within and between groups variance. In addition, is the term "variance" the statistical variance or the sequence differences?

Page 11, Line 262-264: How many haplotypes were found in total? The summation of the number of PHL, Cambodia, Bhutan, Laos, Myanmar, and Vietnam haplotypes was 74, but the sentence on Line 264 stated that "these sequences collapsed when pooled from 76 to 57 haplotypes.

Page 11, Line 265: Would the 26 haplotypes be found in 26 samples? If this is correct, should "a single sequence" be changed to "a single sample"?

Page 12, Line 285: The (Figs 2 and 3) should be changed to (Fig. 3 and 4).

Page 14, Line 335-337: Why would the presence of unique haplotypes, such as PHL21, in Philippine wild pigs suggest the presence of new subspecies of wild pigs? I think carrying the distinct mtDNA haplotype defined based on partial D-loop sequence is insufficient to suggest that the sample is the new subspecies.

Page 16, Line 409-412: What was the result supporting the statement "The distribution of the haplotype frequencies did not indicate equilibrium."? In addition, would the genetic connection to the continental pigs shown by phylogeny and haplotype network suggest the multiple ancestors of Philippine domestic pigs?

Page 18, Line 447-449: The citation should be added to the sentence "This observation is consistent with the situation in domestic pigs in the MSEA, where the mtDNA of European pigs has a negligible impact on the maternal origin of domestic pigs."

Page 20, Line 494-497: Due to the limited sequences available and included in this study, do you have enough evidence to suggest that the Philippine wild pigs were descended from the indigenous wild pigs?

Page 47: Would it be possible to expand the network on Fig. 4, especially for haplogroup D2 and D7, to present the connection between haplotypes?

7. PLOS authors have the option to publish the peer review history of their article (what does this mean?). If published, this will include your full peer review and any attached files.

Reviewer #1: No

Reviewer #4: No

---

## [Author Response · Author response to Decision Letter 1]

23 Jan 2022

First and foremost, we appreciate for considering our manuscript for publication and to the reviewers for their valuable comments and suggestions for improvement. The reviewers' comments indeed helped us to improve the readability and quality of the manuscript. Thank you so much.

Reviewer #1: The subject revised manuscript reports on a study evaluating the origin, dispersal, and level of genetic introgression in Philippine domestic pigs using mitochondrial DNA D-loop analysis. These genetic data are then compared to comparable mtDNA data from domestic pigs and Eurasian wild boar. This revision is much improved over the earlier draft. Again, to my knowledge, this is the first study that has specifically looked at genetic composition as related to the anthropogenic origin of domestic pigs in the Philippines. As such, the data encompassed by the subject manuscript are both unique and important. Overall, the manuscript is well written. The editorial comments noted below are largely intended as suggested changes and should be used by the authors as deemed appropriate:

• Lines 43-44 – Suggest changing “the history of pig dispersal” to “the history of domestic pig (Sus scrofa) dispersal”.

 Revised accordingly.

• Line 45 – Suggest changing “associated with some human migration events” to “associated with human migration events

 Revised accordingly.

• Line 47 – Suggest changing “wild boars” to “wild boar”

 Revised accordingly. Please see Lines 47-48 in the revised manuscript.

• Line 55 – Suggest changing “Philippine pig haplotypes” to “Philippine domestic pig haplotypes”

 Revised accordingly.

• Line 67 – Suggest changing “The wild boar (Sus scrofa L.)” to “The Eurasian wild boar (Sus scrofa L.)”

 Revised accordingly.

• Lines 86-87 – Suggest changing “an exceptionally high levels of endemism” to “an exceptionally high level of endemism”

 Revised accordingly.

• Line 87 – Suggest changing “The Philippines has repeatedly” to “The Philippines have repeatedly”

 Revised accordingly.

• Line 89 - Suggest changing “four endemic wild pigs such as the Philippine” to “four endemic wild pigs including the Philippine”

 Revised accordingly.

• Line 91 – Suggest changing “as well as one native shared with Sundaic” to “as well as one native species shared with the Sundaic”

 Revised accordingly.

• Line 94 – Suggest changing “ubiquitous wild boar S. scrofa was not considered native” to “ubiquitous Eurasian wild boar S. scrofa is not native”

 Revised accordingly.

• Lines 111-112 – Suggest changing “For this reason, [27] excluded” to “For this reason, Larson et al. [27] excluded”

 Revised accordingly. 

• Line 128 – Suggest changing “an introduced pig breed” to “an introduced domestic pig breed”

 Revised accordingly.

• Line 133 - Suggest changing “diversity and structure exhibits a signature” to “diversity and structure exhibit a signature”

 Revised accordingly. 

• Line 217 - Suggest changing “described by [4] with six clades” to “described by Larson et al. [4] with six clades”

 Revised accordingly. Please see Lines 216-217 in the revised manuscript.

• Line 218 – Suggest changing “proposed by [10] were used” to “proposed by Tanaka et al. [10] were used”

 Revised accordingly. Please see Line 217 in the revised manuscript.

• Line 287 – Suggest changing “most pigs in the populations studies” to “most pigs in the population studies”

 Revised accordingly.

• Line 320 - Suggest changing “D7 haplogroup which accounted” to “D7 haplogroup, which accounted”

 Revised accordingly.

• Line 324 – Suggest changing “what [5] recognized” to “what Scandura et al. [5] recognized”

 Revised accordingly. 

• Lines 333-334 – Suggest changing “the previous researches.” to “the previous research.”

 Revised accordingly. See Line 334 in the revised manuscript.

• Line 340 – Suggest changing “wild boars” to “wild boar”

 Revised accordingly. Please see Line 339 in the revised manuscript.

• Line 361 - Suggest changing “contrast, domestic pigs in MSEA combined revealed” to “contrast, combined domestic pigs in MSEA revealed”

 Revised accordingly. Please see Line 360 in the revised manuscript.

• Line 404 - Suggest changing “Philippine pigs” to “Philippine domestic pigs”

 Revised accordingly. Please see Line 403 in the revised manuscript.

• Line 437 - Suggest changing “Before the European arrived in the Philippines” to “Before the Europeans arrived in the Philippines”

 Revised accordingly. Please see Line 436 in the revised manuscript. 

• Line 442 – Suggest changing “which exhibited” to “which is exhibited”

 Revised accordingly. Please see Line 441 in the revised manuscript. 

• Line 446 - Suggest changing “was minimal, which accounted for only” to “was minimal, accounting for only”

 Revised accordingly. Please see Line 445 in the revised manuscript.

• Line 464 - Suggest changing “Chinese pigs, therefore this precluded the origin of these haplogroup” to “Chinese pigs, which precludes the origin of this haplogroup”

 Revised accordingly. Please see Line 463 in the revised manuscript.

• Line 480 – Suggest changing “populations despite shared haplotypes could” to “populations, despite shared haplotypes, could”

 Revised accordingly. Please see Line 479 in the revised manuscript.

• Lines 490-497 – What “Philippine wild pigs” are being referred to here? The Philippine wild pigs that Bill Oliver was referring to in the IBEX article cited are not Sus scrofa, and in fact are all separate species from either domestic or wild Sus scrofa. Without identifying the species of the Philippine wild pigs that you are referring to, I’m not sure of the value of this paragraph to the rest of the study.

 Revised accordingly. This paragraph was deleted. 

• Lines 510-511 - Suggest changing “In addition, [72] have previously discussed” to “In addition, Scandura et al. [72] have previously discussed”

 Revised accordingly. Please see Lines 500-501 in the revised manuscript. 

• Line 525 – Suggest changing “distributions which represent” to “distributions that represent”

 Revised accordingly. Please see Line 515 in the revised manuscript.

• Line 560 - Suggest changing “reported by [54]” to “reported by Frantz et al. [54]”

 Revised accordingly. Please see Line 550 in the revised manuscript.

• Line 566 – Suggest changing “exchange of pigs” to “exchange of domestic pigs”

 Revised accordingly. Please see Line 558 in the revised manuscript.

• Line 574 – What is the name “native pig populations” referring to here, domestic or wild pigs? There are no “native” domestic pig populations in the Philippines.

 Revised accordingly. We change “native” to “local”. Please see Line 567 in the revised manuscript.

Reviewer #4: This manuscript aimed to present the origin and distribution of Philippine domestic pigs based on partial mitochondrial D-loop sequences analysis. The results showed the genetic connection between the Philippine domestic and continental pigs, supporting the Neolithic-Austronesian expansion model and proposing the possible alternative migration route. The data and analyses presented in the manuscript are appropriate, and the results support the key messages. The sequences under accessions MN625805-MN625830 are searchable; however, the sequences under accessions MW924902-MW92973 could not be found in the GenBank database. I am reviewing another manuscript submitted to a different journal. In that manuscript, some pictures of pigs looked the same as all sub-figures in Fig. 2, except Fig. 2F. In addition, some contents in the introduction section of that manuscript were very similar, if not the same, as some parts of the introduction of this manuscript; for example, the content on Page 4, Line 96-103 starts with the words "To date." However, more Philippine pig samples were included, and some analyses applied on mitochondrial D-loop sequences in that manuscript differed from this manuscript. In addition, the conclusion in that manuscript could be considered as an answer of the future direction of this manuscript mentioned on Page 23, Line 567-569, that " a more thorough investigation should be conducted to relate the possible ancestral diffusion that occurred from MSEA to the Philippines via Sundaland. 

 The authors highly appreciate the positive comments and suggestions by the Reviewer and for considering our manuscript for publication. With regards to the accessibility of sequences in the GenBank database, they are still being processed and will be released to the public once finalized. Submissions are not automatically deposited into the GenBank after being accessioned, ensuring first that it is free of errors or problems. This manuscript was submitted in the bioRxiv (a preprint server) to accelerate the dissemination of our research. This submission makes our work citable even before the formal publication. Thus, this article has already the Digital Object Identifier (DOI) that will later link the preprint and the final version of the manuscript when it is officially published, including citations. As we all know, peer-review takes time and getting submission to publication can take several months or even a year. Moreover, as our Laboratory handles huge research projects which in any circumstance be published ahead of time, this article is well-cited and credited in all our successive submissions. As also stressed by some Reviewers, this is the first study that has specifically looked at genetic composition relating to the anthropogenic origin of domestic pigs in the Philippines. This study is considered a baseline study and, therefore, a hot topic in the field. However, this manuscript still has an underlying question that needs to be addressed (e.g., stressed on Page 23, Lines 567-569). The other manuscript that we recently submitted is a confirmatory and a continuation study about this paper (as likewise, the Reviewer mentioned in his comments), which allows us to address the limitations of this article on that manuscript. Please see the new Figure 2 in the revised manuscript. Thank you very much for this clarification and the authors appreciate the Reviewer.

Despite the concerns in the previous paragraph, the following comments should be addressed before accepting publication. 

Page 2, Line 61-62: Regarding the sentence on these lines, what is the population showing sudden decline? The population name, such as Philippine pigs, Mainland Southeast Asian pigs, should be specified. In my opinion, both Bayesian skyline plots (Fig. 7) did not clearly show a sudden decline, so I am wondering what result supported this statement? This comment also applied to the statement "The sudden population decline occurred approximately 25,000 years before present (BP)" on Page 15, Line 377-378 and "The underlying sudden population decline as predicted in the BSP markedly followed the LGM period" on Page 23, Line 571-572.

 Revised accordingly. The word “Philippine domestic pigs” was added to these sentences. The authors are pointing out the BSP for Philippine samples. As the Figure projects, the BSP trend showed a stationary period of maternal effective population size however, the decline trend can be observed approximately 25,000 years BP which compliments the multimodal mismatch distribution of the Philippine samples. However, after further deliberation by all the authors and to avoid confusion, we decided to delete the word “sudden” in these sentences. 

Page 8-9, Line 214-216: The statement "This method calculates the net divergence of each taxon from all other taxa as the sum of the individual distances from variance within and among groups of Philippine pigs and comparison sequences." should be revised. It was not whether the net divergence of each taxon was the sum of the individual distance, and the distance was calculated from within and between groups variance. In addition, is the term "variance" the statistical variance or the sequence differences?

 Revised accordingly. Revised to “This method calculates the net divergence of each taxon from all other taxa as the sum of the individual distances from variance within and among groups.” We are referring the statistical variance. Please see Lines 214-216 in the revised manuscript.

Page 11, Line 262-264: How many haplotypes were found in total? The summation of the number of PHL, Cambodia, Bhutan, Laos, Myanmar, and Vietnam haplotypes was 74, but the sentence on Line 264 stated that "these sequences collapsed when pooled from 76 to 57 haplotypes.

Page 11, Line 265: Would the 26 haplotypes be found in 26 samples? If this is correct, should "a single sequence" be changed to "a single sample"?

 Revised accordingly. Please see Lines 264- 265 in the revised manuscript.

Page 12, Line 285: The (Figs 2 and 3) should be changed to (Fig. 3 and 4).

 Revised accordingly. Please see Line 285 in the revised manuscript.

Page 14, Line 335-337: Why would the presence of unique haplotypes, such as PHL21, in Philippine wild pigs suggest the presence of new subspecies of wild pigs? I think carrying the distinct mtDNA haplotype defined based on partial D-loop sequence is insufficient to suggest that the sample is the new subspecies.

 Revised accordingly. This sentence was deleted, as likewise suggested by the first Reviewer. Thank you very much.

Page 16, Line 409-412: What was the result supporting the statement "The distribution of the haplotype frequencies did not indicate equilibrium."? In addition, would the genetic connection to the continental pigs shown by phylogeny and haplotype network suggest the multiple ancestors of Philippine domestic pigs?

 The authors referred to the distribution of the shared haplotypes that are not equal. As based on the distribution, there are some haplotypes that are inter-island and regionally shared. For question number 2, we based this statement as clearly indicated by literatures that Eurasian wild boar is not native to the Philippine. Further, our analysis of the contemporary samples showed closed affinity and similarity in the mutational motif to those continental wild and domestic pigs, as well as where the fossil records where the major domestication took place.

Page 18, Line 447-449: The citation should be added to the sentence "This observation is consistent with the situation in domestic pigs in the MSEA, where the mtDNA of European pigs has a negligible impact on the maternal origin of domestic pigs."

 Revised accordingly. Please see Line 447 in the revised manuscript.

Page 20, Line 494-497: Due to the limited sequences available and included in this study, do you have enough evidence to suggest that the Philippine wild pigs were descended from the indigenous wild pigs?

 Revised accordingly. This paragraph was deleted as also suggested by Reviewer 1.

Page 47: Would it be possible to expand the network on Fig. 4, especially for haplogroup D2 and D7, to present the connection between haplotypes?

 Revised accordingly. Please see Figure 4.

---

## [Editor Report · Decision Letter 2]

2 Feb 2022

PONE-D-21-18979R2Insights on the historical biogeography of Philippine domestic pigs and its relationship with continental domestic pigs and wild boarsPLOS ONE

Dear Dr. Layos,

Thank you for submitting your manuscript to PLOS ONE. After careful consideration of the revised manuscript, we feel that it has merit; however, some additional revisions are needed to meet PLOS ONE’s publication criteria. We invite you to submit a revised version of the manuscript that addresses my comments listed below. Line 55:  "collapsed with" is unclear. If the intent is to highlight the prevalence of D7 haplogroup in the Philippines perhaps "included" would be more appropriate. Lines 63-64: Which finding highlights an important challenge managing Philippine pig genetic resources and what is the challenge? Lines 202-204: The description of Arlequin AMOVA fixation indices is inaccurate and needs to be revised. As noted in the Arlequin user manual (Excoffier et al. 2005), PhiST tests for deviations from panmixia within the full sample, PhiCT tests the significance of designated population groupings, and PhiSC tests for differences among populations within groups.Excoffier, L., Laval, G., & Schneider, S. (2005). Arlequin (version 3.0): an integrated software package for population genetics data analysis. *Evolutionary bioinformatics*, *1*, 117693430500100003.Lines 220-252: My feeling is that population demographic analyses as presented may be inaccurate because groupings include historically isolated populations. This violates model assumptions and can bias results by reflecting changes in the degree of structure rather than changes in population size (eg. Heller et al. 2013). Given this, I think the impact of the paper could be improved by leaving out demographic analyses and placing the focus more clearly on phylogenetic and phylogeographic results. However, if you decide to retain demographic analyses please include a caveat highlighting the potential impact of historic population subdivision on demographic estimates.Heller, R., Chikhi, L., & Siegismund, H. R. (2013). The confounding effect of population structure on Bayesian skyline plot inferences of demographic history. *PloS one*, *8*(5), e62992.Line 264-265: The sentence starting with "Twenty-six..." is unclear. Line 284: Because PhiST reflects the average contribution of population subdivision to deviation from panmixia, significance is possible even if some populations comparisons are not significantly different. Given this, please change "at all geographic locations examined" to something like "indicating significant genetic structure among the geographic locations examined."Table 2: Please revise the "source of variation" column and related notes below the table to more accurately reflect Arlequin fixation indices. Specifically, PhiSC is more accurately described as "among populations within groups". Table 3: Is pairwise Fst or PhiST presented here? If genetic distance between haplotypes was included in the analysis then PhiST would be correct.Line 333: "nomenclature quotation" is unclear. Line 507: Is "Fst" a typo? If not, some clarification in the methods is needed. Line 507: "individuals" seems to be a typo. Please correct.Figure S1: There are no indels so the the definition of the "minus" symbol can be deletedFigure S1: D7 is not included in the figure. 

Figure 1: Please enlarge the haplotype key to ensure readability.Figure 5-7: If demographic analyses are retained these figures should be moved to Supplemental data.

We look forward to receiving your revised manuscript.

Kind regards,

Jeffrey A. Eble, Ph.D.

Academic Editor

PLOS ONE
---

## [Author Response · Author response to Decision Letter 2]

5 Feb 2022

Line 55: "collapsed with" is unclear. If the intent is to highlight the prevalence of D7 haplogroup in the Philippines perhaps "included" would be more appropriate. 

Revised accordingly. 

Lines 63-64: Which finding highlights an important challenge managing Philippine pig genetic resources and what is the challenge? 

One of the reviewers recommended to add concluding statement in the abstract. However, to be more specific, we have changed this sentence to “The results of this study will support the conservation strategies and improvements of economically important genetic resources in the Philippines.”

Lines 202-204: The description of Arlequin AMOVA fixation indices is inaccurate and needs to be revised. As noted in the Arlequin user manual (Excoffier et al. 2005), PhiST tests for deviations from panmixia within the full sample, PhiCT tests the significance of designated population groupings, and PhiSC tests for differences among populations within groups.

Excoffier, L., Laval, G., & Schneider, S. (2005). Arlequin (version 3.0): an integrated software package for population genetics data analysis. Evolutionary bioinformatics, 1, 117693430500100003.

Revised accordingly. Please see Lines 200-201 in the revised manuscript.

Lines 220-252: My feeling is that population demographic analyses as presented may be inaccurate because groupings include historically isolated populations. This violates model assumptions and can bias results by reflecting changes in the degree of structure rather than changes in population size (eg. Heller et al. 2013). Given this, I think the impact of the paper could be improved by leaving out demographic analyses and placing the focus more clearly on phylogenetic and phylogeographic results. However, if you decide to retain demographic analyses please include a caveat highlighting the potential impact of historic population subdivision on demographic estimates.

Heller, R., Chikhi, L., & Siegismund, H. R. (2013). The confounding effect of population structure on Bayesian skyline plot inferences of demographic history. PloS one, 8(5), e62992.

Revised accordingly. We have decided to retain the demographic analyses by adding caution on the impact of historic population subdivision on demographic estimates for the Philippine samples. We added the following sentences, “However, since the Philippine samples included historically isolated populations that may have caused a sound impact on the demographic estimates, it could also be that the result reflected here is merely the changes in the degree of structure rather than changes in the population size.” Please see Lines 542-545 in the revised manuscript.

The following citation was also added:

Heller R, Chikhi L, Siegismund HR. The confounding effect of population structure on Bayesian Skyline Plot inferences of demographic history. PLoS ONE. 2013;8(5): e62992.

Line 264-265: The sentence starting with "Twenty-six..." is unclear. 

Revised accordingly. We changed to “Twenty-six (6 PH and 20 MSEA) of the 57 haplotypes were found only once among the sequences.”. Please see Lines 266-267 in the revised manuscript.

Line 284: Because PhiST reflects the average contribution of population subdivision to deviation from panmixia, significance is possible even if some populations comparisons are not significantly different. Given this, please change "at all geographic locations examined" to something like "indicating significant genetic structure among the geographic locations examined."

Revised accordingly. Please see Line 285 in the revised manuscript.

Table 2: Please revise the "source of variation" column and related notes below the table to more accurately reflect Arlequin fixation indices. Specifically, PhiSC is more accurately described as "among populations within groups". 

Revised accordingly. 

Table 3: Is pairwise Fst or PhiST presented here? If genetic distance between haplotypes was included in the analysis then PhiST would be correct.

The population pairwise FST is the one presented in Table 3 and not the PhiST. The authors, however, acknowledged that our discussion on the methodology was confusing. Please check the revised methodology, RE: Phylogenetic and population structure analysis. Lines 195-206 in the revised manuscript.

Line 333: "nomenclature quotation" is unclear. 

Revised accordingly. Please see Line 341 in the revised manuscript.

Line 507: Is "Fst" a typo? If not, some clarification in the methods is needed. 

Our apology for the confusion. This is not a typo. Please check the revised methodology, RE: Phylogenetic and population structure analysis.

Line 507: "individuals" seems to be a typo. Please correct.

Revised accordingly. Changed from “individuals” to “population”. Please see Line 500 in the revised manuscript.

Figure S1: There are no indels so the definition of the "minus" symbol can be deleted

Revised accordingly.

Figure S1: D7 is not included in the figure. 

Revised accordingly. Please see the revised Figure S1.

Figure 1: Please enlarge the haplotype key to ensure readability.

Revised accordingly. Please see the revised Figure 1.

Figure 5-7: If demographic analyses are retained these figures should be moved to Supplemental data.

Revised accordingly. These Figures were moved to the supplemental data.

---

## [Editor Report · Decision Letter 3]

28 Feb 2022

PONE-D-21-18979R3Insights on the historical biogeography of Philippine domestic pigs and its relationship with continental domestic pigs and wild boarsPLOS ONE

Dear Dr. Layos,

Thank you for submitting your manuscript to PLOS ONE. We very much appreciate your careful attention to reviewer and editor comments throughout the review process; however, we feel one additional revision with a focus on grammar and punctuation is needed to fully meet PLOS ONE’s publication criteria. Therefore, we invite you to submit a revised version of the manuscript that addresses the points listed below and includes a final close review of the manuscript for grammatical and formatting errors. 

Line 48- delete "a" in "revealed a considerable"Line 50- add comma between "analysis with" AND delete comma between "(5.10%), harboringLine 55- delete "with"Line 59-61- This sentence is unclear and should be revised.Line 132- delete "even"Line 202- delete "namely"Line 212- what was Fst used for pairwise analyses while PHIst was used for AMOVA? If a mutational model was included in pairwise comparisons then PHIst should be used. If genetic distance was not included then Fst is the accurate parameter, but if that is the case then I think a brief justification is needed to help readers understand why genetic distance was included in one set of analyses and not the other.Line 297- is "(Fst values, p<0.01)" a typo? Did you mean to include the range of Fst values here?Line 336- delete "A" in "A widespread Asian ancestry"Line 422- "underlies as a genetic basis" is unclear. Please reviseLine 427- what "equilibrium" are you referencing?Lines 435-437- Please revise for clarity and grammar. For example... "Thus, these events may have paved the way for the introduction of multiple distinct lineages of domestic pigs to the Philippines, as has been documented for chickens [56] and goats [57]."Lines 443-446- Please revise for clarity and grammar. Line 446- replace "the preliminary studies that revealed" with "preliminary genetic studies revealed"Line 449- add "in the Philippines" after "D2 haplotypes"Line 458- change "precisely evident as shown by" to "supported by"Line 461-  change "a crucial" to "an important"Lines 471-474- This sentence is unclear, please revise.Line 500- replace "and the consequent" with ", and subsequentLines 500-503- This sentence is unclear, please revise.Line 503- is "founder" a typo? If I understand your point then "founder event" would be more accurate.Line 504- "population" should be pluralLine 568- "sound impact" is unclear, please revise.Line 569- delete "the" before "changes"Line 570- delete "the" before "population size"Line 578-580- This sentence is unclear, please revise.Line 583- delete "flow"

We look forward to receiving your revised manuscript.

Kind regards,

Jeffrey A. Eble, Ph.D.

Academic Editor

PLOS ONE
---

## [Author Response · Author response to Decision Letter 3]

8 Mar 2022

Thank you for considering our manuscript for publication and the comments and suggestions for improvement. 

Line 48- delete "a" in "revealed a considerable"

 Revised accordingly.

Line 50- add comma between "analysis with" AND delete comma between "(5.10%), harboring

 Revised accordingly.

Line 55- delete "with"

 Revised accordingly.

Line 59-61- This sentence is unclear and should be revised.

 Revised accordingly. Revised to “The analyses of mismatch distribution and neutrality test were consistent with the Bayesian skyline plot which showed a long stationary period of effective population size.”

Line 132- delete "even"

 Revised accordingly. Please see Line 128 in the revised manuscript.

Line 202- delete "namely"

 Revised accordingly. Please see Line 198 in the revised manuscript. 

Line 212- what was Fst used for pairwise analyses while PHIst was used for AMOVA? If a mutational model was included in pairwise comparisons then PHIst should be used. If genetic distance was not included then Fst is the accurate parameter, but if that is the case then I think a brief justification is needed to help readers understand why genetic distance was included in one set of analyses and not the other.

 Our apology for the confusion. Unlike in AMOVA, where we calculated the standard “variance” components of each hierarchic characteristic at 3 levels: among groups defined a priori, among populations within groups, and among localities within populations (Excoffier et al., 1992), the population pairwise FST, on the other hand was performed to estimate the overall genetic divergence between two populations (i.e., BTN vs. KHM; VNM vs. PHL; etc.). The AMOVA here was done only to estimate population differentiation of the geographic divisions of the four genetic structure hypotheses. 

Line 297- is "(Fst values, p<0.01)" a typo? Did you mean to include the range of Fst values here?

 Yes, this is a typo. Revised accordingly. Please see Line 289 in the revised manuscript. 

Line 336- delete "A" in "A widespread Asian ancestry"

 Revised accordingly. Please see Line 322 in the revised manuscript.

Line 422- "underlies as a genetic basis" is unclear. Please revise

 Revised accordingly. Please see Line 393 in the revised manuscript.

Line 427- what "equilibrium" are you referencing?

 The authors referred to the distribution of the shared haplotypes which are not equal. As based on the distribution, there are some haplotypes that are inter-island and regionally shared.

Lines 435-437- Please revise for clarity and grammar. For example... "Thus, these events may have paved the way for the introduction of multiple distinct lineages of domestic pigs to the Philippines, as has been documented for chickens [56] and goats [57]."

 Revised accordingly. Please see Lines 407-408 in the revised manuscript.

Lines 443-446- Please revise for clarity and grammar. 

 Revised accordingly. Revised to “The complexity surrounding the Neolithic Austronesian expansion and dispersal has recently raised several questions due to the limited genetic studies conducted in the Philippine archipelago, as the faunal assemblages in this key region have provided some clues [4].” Please see Lines 414 to 416 in the revised manuscript. 

Line 446- replace "the preliminary studies that revealed" with "preliminary genetic studies revealed"

 Revised accordingly. Please see Line 417 in the revised manuscript.

Line 449- add "in the Philippines" after "D2 haplotypes"

 Revised accordingly. Please see Line 421 in the revised manuscript. 

Line 458- change "precisely evident as shown by" to "supported by"

 Revised accordingly. Please see Line 429 in the revised manuscript.

Line 461- change "a crucial" to "an important"

 Revised accordingly. Please see Line 431-432 in the revise manuscript. 

Lines 471-474- This sentence is unclear, please revise.

 Revised accordingly. Revised to “. Previously, it was emphasized that European pigs possessed both Asian and European mtDNA resulting from the extensive history of interbreeding with predominantly Asian mtDNA introgression [62-63,3], which currently accounts for approximately 20-35% Asian contribution to western modern breeds [64-66].” Please see Lines 442-445 in the revised manuscript.

Line 500- replace "and the consequent" with ", and subsequent

 Revised accordingly. Please see Line 470-471 in the revised manuscript.

Lines 500-503- This sentence is unclear, please revise.

 Revised accordingly. Rephrased to “In addition, as predicted by the theory of genetic isolation by distance, population differentiation usually occurs when there is migration of a certain population away from its founder population, leading to a reduction in genetic diversity.” Please see Lines 471-475 in the revised manuscript.

Line 503- is "founder" a typo? If I understand your point then "founder event" would be more accurate.

 Revised accordingly. Please see Line 475 in the revised manuscript.

Line 504- "population" should be plural

 Revised accordingly. Please see Line 475 in the revised manuscript. 

Line 568- "sound impact" is unclear, please revise.

 Revised accordingly. Please see Line 539 in the revised manuscript.

Line 569- delete "the" before "changes"

 Revised accordingly. Please see Line 540 in the revised manuscript.

Line 570- delete "the" before "population size"

 Revised accordingly. Please see Line 541 in the revised manuscript.

Line 578-580- This sentence is unclear, please revise.

 Revised accordingly. Revised to “This study provided important insights that will properly help address the contradicting hypothesis of a possible human-mediated translocation and exchange of domestic pigs in the Philippines.” Please see Lines 549-551 in the revised manuscript.

Line 583- delete "flow"

 Revised accordingly. Please see Line 554 in the revised manuscript.

---

## [Editor Report · Decision Letter 4]

15 Mar 2022

Insights on the historical biogeography of Philippine domestic pigs and its relationship with continental domestic pigs and wild boars

PONE-D-21-18979R4

Dear Dr. Layos,

We’re pleased to inform you that your manuscript has been judged scientifically suitable for publication and will be formally accepted for publication once it meets all outstanding technical requirements.

Kind regards,

Jeffrey A. Eble, Ph.D.

Academic Editor

PLOS ONE
---

## [Editor Report · Acceptance letter]

17 Mar 2022

PONE-D-21-18979R4 

Insights on the historical biogeography of Philippine domestic pigs and
its relationship with continental domestic pigs and wild boars 

Dear Dr. Layos:

I'm pleased to inform you that your manuscript has been deemed suitable for publication in PLOS ONE. Congratulations! Your manuscript is now with our production department. 

Kind regards, 

on behalf of

Dr. Jeffrey A. Eble 

Academic Editor

PLOS ONE